# A Survey on Applications of H-Technique: Revisiting Security Analysis of PRP and PRF

**DOI:** 10.3390/e24040462

**Published:** 2022-03-26

**Authors:** Ashwin Jha, Mridul Nandi

**Affiliations:** 1CISPA Helmholtz Center for Information Security, 66123 Saarbrücken, Germany; 2Indian Statistical Institute, Kolkata 700108, India; mridul@isical.ac.in

**Keywords:** provable security, Coefficients H technique, Feistel, ENR, LDT, HCTR, TET

## Abstract

The Coefficients H technique (also called the H-technique), developed by Patarin circa 1991, is a tool used to obtain the upper bounds on distinguishing advantages. This tool is known to provide relatively simple and (in some cases) tight bound proofs in comparison to some other well-known tools, such as the game-playing technique and random systems methodology. In this systematization of knowledge (SoK) paper, we aim to provide a brief survey on the H-technique. The SoK is presented in four parts. First, we redevelop the necessary nomenclature and tools required to study the security of any symmetric-key design, especially in the H-technique setting. Second, we provide a full description of the H-technique and some related tools. Third, we present (simple) H-technique-based proofs for some popular symmetric-key designs, across different paradigms. Finally, we show that the H-technique can actually provide optimal bounds on distinguishing advantages.

## 1. Introduction

The general goal of any cryptographic scheme is to achieve some kind of indistinguishability (pseudorandom behavior) from an ideal (random) system. In this respect, distinguishing games have a key role in defining cryptographic security definitions. In symmetric-key cryptography, pseudorandom functions or PRFs [1] and (strong) pseudorandom permutations or (S)PRPs [2] have been defined via distinguishing games. Informally, an adversary interacts with either the keyed construction (real system) of interest or with an ideal system, such as a uniform random function or permutation. The adversary’s goal is to reliably distinguish which one it is interacting with.If no adversary can distinguish the real system from the ideal system with a non-negligible probability, we say that the construction is *pseudo-ideal* (e.g., PRF or PRP).

Upon closer inspection of the security proofs for most of the symmetric-key designs, we mostly see that the underlying primitives are first replaced by some ideal primitives. This can be justified using the hybrid argument at the cost of the distinguishing advantages of each of the underlying primitives. Once we replace these underlying primitives with ideal candidates, we obtain the so-called *hybrid* or *quasi random* construction (information-theoretically indistinguishable from ideal candidates). The next and final step is to provide a security analysis of the hybrid construction. This is performed in a purely information-theoretic setting. So, in a way, the provable security analysis guarantees the security of the construction if the underlying primitives are indistinguishable from their ideal counterparts. In this paper we focus on the security analysis of such hybrid constructions.

### 1.1. Revisiting Some Popular Proof Techniques

Provable securityresults in symmetric-key can be broadly classified according to the proof techniques used. Different constructions may warrant different proof techniques depending upon the proof’s complexity, the desired security bound, and in some cases, the author’s bias.We briefly discuss some of these techniques.

Game-Playing Technique: Arguably the most popular, and certainly the oldest proof technique is the so-called game-playing technique [3,4]. At a high level the proofs based on this technique use a sequence of games, in which each game is an interaction between the adversary and an oracle. The proof starts with the game corresponding to the real construction and proceeds toward the game corresponding to the ideal system by making stepwise transitions to some intermediate games. Each transition may enable the adversary to gain some advantage, and the cumulative advantage of all such transitions determines the security bound. This high-level view of the game-playing technique is clearly demonstrated in several early works [5,6]. In more recent years, Shoup used this technique extensively [4,7,8,9,10,11].

The contemporary version of this tool is derived from Bellare and Rogaway’s systematized treatment [3], called the *code-based game-playing* technique. In this approach, the games are written in pseudocode language, each having their own internal variables and flags. Two games are said to be identical if they are syntactically identical. Usually the syntactical identity breaks when one of the game sets a Boolean flag *bad* to true. Consequently, the adversary’s distinguishing advantage in any game is upper-bounded by the probability that this flag will be set. The game-playing technique has been used to prove the security of almost all types of security notions in symmetric-key cryptography.For example, the game-playing technique was employed in the following:(Tweakable) (S)PRPs such as three and four rounds of Feistel [2], CLRW2 [12], etc.;PRFs and MACs such as CBC-MAC [13,14], ECBC, FCBC and XCBC [15], PMAC+ [16], sum of ECBC [17], etc.;(Tweakable) enciphering schemes such as CMC [18], EME [19], TET [20], HCH [21], HCTR [22], XLS [23], HEH [24,25], etc.;Online ciphers such as HCBC1 and HCBC2 [26], TC1, TC2 and TC3 [27], POEx [28], XTC [29], etc.;AE schemes such as SIV [30], OCB [31,32,33], COPA [34], POET [35], etc.

Coefficients H Technique: Patarin formally introduced the Coefficients H technique [36] at SAC 2008, although the technique had already been used in some of his earlier works [37,38,39,40,41]. In fact, it was Vaudenay who first reported the H-technique publicly in his decorrelation theory [42]. However, he mentioned that the technique was described in Patarin’s PhD thesis [38], written in French. Independently, Bernstein rediscovered a similar variant of the result in [43], referred to as the *interpolation theorem*. This was later strengthened by Nandi [44] as the *strong interpolation theorem*. Later, Chen and Steinberger presented a renewed interpretation of the H-technique in their work on key alternating ciphers [45]. They expressed the hope that their “*paper will serve as a useful additional tutorial on (or introduction to) Patarin’s H-coefficient technique, which still seems to suffer from a lack of exposure*”. This modernization indeed popularized the H-technique, as to the best of our knowledge, all the recent applications consider this renewed description of the H-technique. We remark that Mennink’s uses of the H-technique in [46,47] offer a relatively simple yet similar exposition of the technique with relatively simple constructions.

At a very high level, the H-technique concentrates on the input-output tuple generated by an adversary’s interaction with the oracle at hand, called the transcript. In the simplest case, the H-technique states that the distinguishing advantage is bounded by one minus a lower bound of the ratio of the probability that an *attainable* transcript can be realized by the real oracle to the probability that it can be realized by the ideal oracle. A transcript is called attainable if the probability that it can be realized by the ideal oracle is non-zero.

For example, suppose an adversary A wants to distinguish a uniform random function ρ:D→D (the real oracle) from a uniform random permutation π:D→D (the ideal oracle) by making *q* queries to the oracle at hand (this is popularly known as PRP-PRF switching lemma [3]). A typical attainable transcript for this distinguishing game would look like $\omega =(\left(x_{1},y_{1}\right),\left(x_{2},y_{2}\right),\ldots ,\left(x_{q},y_{q}\right))$, where $x_{i}$ and $y_{i}$ denote the *i*-th query and response, respectively. For an attainable transcript for the uniform random permutation, we must have $x_{i}=x_{j}\Leftrightarrow y_{i}=y_{j}$ for all i≠j. Without the loss of generality, we may assume that $x_{i}\neq x_{j}$, as A does not attain any benefits from duplicate queries. Let $\Theta_{0}$ and $\Theta_{1}$ be the transcript random variable generated by A’s interaction with π and ρ, respectively. Then, it is easy to see that
$$\mathrm{Pr}\left[\Theta_{0}=\omega \right]={\mathrm{Pr}}_{\pi }\left[\pi \left(x_{1}\right)=y_{1},\ldots ,\pi \left(x_{q}\right)=y_{q}\right]=\frac{1}{2^{n}\left(2^{n}-1\right)\cdots \left(2^{n}-q+1\right)},$$
and
$$\mathrm{Pr}\left[\Theta_{1}=\omega \right]={\mathrm{Pr}}_{\rho }\left[\rho \left(x_{1}\right)=y_{1},\ldots ,\rho \left(x_{q}\right)=y_{q}\right]=\frac{1}{2^{nq}}.$$
Thus, the ratio of the above two probabilities is lower-bounded as
$$\frac{\mathrm{Pr}[\Theta_{0}=\omega ]}{\mathrm{Pr}[\Theta_{1}=\omega ]}=\frac{2^{n}\left(2^{n}-1\right)\cdots \left(2^{n}-q+1\right)}{2^{nq}}\geq \left(1-\frac{q(q-1)}{2^{n+1}}\right).$$
Finally, the Coefficients H technique states that A’s advantage in distinguishing ρ from π is upper-bounded by $q^{2}/2^{n+1}$.

The above example is quite simple. However, in many cases, it might be possible that certain transcripts are bad (i.e., they may lead to inconsistency or are improbable) in the real or ideal world. In those cases, we also have to add the probability of realizing a bad transcript in the ideal world to the distinguishing advantage. For example, if we interchange the real and the ideal world in the above example, we can define a transcript $\omega =(\left(x_{1},y_{1}\right),\left(x_{2},y_{2}\right),\ldots ,\left(x_{q},y_{q}\right))$ as bad if for some i≠j, $y_{i}=y_{j}$. This can happen with a probability of at most $q^{2}/2^{n+1}$ in the case of the ideal world (which is now the uniform random function). For a good transcript ω (from the above calculation), it is easy to see that $\frac{\mathrm{Pr}[\Theta_{1}=\omega ]}{\mathrm{Pr}[\Theta_{0}=\omega ]}\geq 1$. A variant of H-technique which allows bad transcripts will then give the same bound (which is the sum of probability of a bad transcript in the ideal world and the maximum value of 1 minus the ratio over all good transcripts).In the second approach, the maximum value of 1 minus the ratio is, at most, zero and so it is simply bounded by the bad transcript probability.

Note that we have consciously ignored some key technical details, such as A’s computational resources and its probabilistic nature, in order to give a short overview on how the H-technique tool works. In later sections, we will provide a formal description of the H-technique in full generality. Nevertheless, it can still be observed that the H-technique, unlike the game-playing technique, does not make any implicit assumptions about the probability distribution of the oracles and requires an explicit probability computation in both the worlds to provide a bound for the ratio. In recent years, there has been a steady rise in the number of applications of the H-technique. Some of the schemes which have been analyzed with the H-technique include:(Online) PRP/SPRPs such as Feistel [41,48,49], MHCBC and MCBC [50], (tweakable) even-Mansour [45,51,52,53,54,55], FMix [56], OleF [57], two-round LDT [58], CLRW2 [59], etc.;PRFs and MACs such as CBC-MAC [60], EWCDM [61,62], HaT and NaT [63], 1k-PMAC+ [64], EHtM [65], ZMAC+ [66], DWCDM [67], etc.;AE schemes such as ELmE [68], COFB [69], OCB3 [70], Beetle [71], GCM-SIV [72], etc.

Maurer’s Random Systems Methodology: At Eurocrypt 2002, Maurer introduced the random systems methodology (also called Maurer’s methodology) for indistinguishability proofs [73]. The randoms system methodology defines a sequence of conditional probabilities associated with a system, i.e., the interaction between an adversary and an oracle. It further defines the notion of a monotone binary condition associated to a system. Two systems are said to be equivalent until a monotone binary condition *B*, if they give rise to the same sequence of conditional probabilities while B=0. This formalizes the identical-until-bad philosophy of the game-playing technique. Expectedly, the advantage is bounded by the probability that the monotone condition changes to 1. Note that in contrast to this approach, the H-technique considers the joint probability distribution for the systems. Applications of Maurer’s methodology were first presented in some indistinguishability and composition proofs [73,74,75,76]. Later, Maurer’s methodology was also applied to prove the security of PMAC, TMAC and XCBC [77], ENR and its variants [78,79,80], and XTX [81].

Expectation Method: The expectation method, developed by Hoang and Tessaro [82], is a generalization of the H-technique, in which the expected value of the ratio is used instead of a constant (i.e., independent of the transcript) lower bound. The expectation method has been applied to obtain exact bounds in [82] and to obtain multi-user security in [82,83,84].

Coupling Technique: The coupling technique [51,85,86,87] is a very useful tool for upper-bounding of the distinguishing advantage (mostly for the nonadaptive distinguisher) of iterated structures. For example, it has been applied, among others, to the iterated even-Mansour [51], iterated tweakable even-Mansour [54], and cascaded LRW2 [88] schemes. The high level idea is simple: the coupling lemma is used to bound the statistical distance of *r*-rounds of some iterated scheme from uniform distribution. This step is non-adaptive in nature. Now, given the non-adaptive bound, a straightforward application of the composition lemma from [75] gives adaptive security for 2r rounds of the iterated scheme. The coupling technique is known for having notoriously loose bounds. This is because even if the coupling lemma gives a tight bound for the non-adaptive security of *r* rounds, this does not say anything about the adaptive security of *r* rounds. It might be possible to obtain the desired level of adaptive security at *r* rounds itself, but the technique requires 2r rounds.

$\chi^{2}$-Method and Hellinger Distance: The $\chi^{2}$-method was proposed by Dai, Hoang and Tessaro [89], in which the statistical distance is bounded in terms of the expectation of the conditional $\chi^{2}$-distances. The $\chi^{2}$-method gave improved bounds in some cases, such as the sum of permutations and EDM [89,90], where the H-technique failed. Bhattacharya and Nandi explored the applications of the $\chi^{2}$-method in analyzing the PRF security of a sum of permutation variant [91] and the indifferentiability of a sum of permutations [92]. Recently, beyond-birthday security analysis of three-round LDT [93] has been demonstrated. The $\chi^{2}$ method is quite useful in certain cases in which it is easy to compute the conditional probabilities such as the sum of permutation. However, there is no clear picture as to its utility in cases in which the conditional probability is not that easy to compute, such as hash-based schemes.

The application of a different distance notion for the binding of the statistical distance is not new. In fact, much earlier Steinberger used the Hellinger distance [94] to study the key alternating ciphers. However, this method is yet to be explored for other constructions.

### 1.2. Our Contribution

The contributions of this SoK are fourfold. First, we reformulate an interactive algorithm in its functional view, which provides the language of the proof of symmetric-key designs. Second, we provide a complete description of the H-technique tool, along with some related tools. Third, we revisit the security analysis of some of the well known symmetric-key constructions across different paradigms. Specifically, we give H-technique based proofs for the following constructions:Hash-based schemes: hash-then-PRF, hash-then-TBC [66] and ENR [78,79]. In the case of ENR we study a generic scheme, called ${\mathrm{NR}}^{⋆}$, which allows for simple proofs for both ENR and LDT [58].Feistel cipher: the three-round Luby–Rackoff [2] and the three-round tweakable block-cipher-based Luby–Rackoff scheme [95].SPRP enciphering schemes: HCTR [22] and TET [20].Beyond-birthday-bound secure PRFs: SoP [96,97,98] and SoEM22 [99].

Finally, we show that the extended version of the H-technique can achieve optimal security bounds. As a side result, we provide an alternate proof for the composition of non-adaptive PRPs [75].

The above given constructions were chosen for varied reasons. In some cases, we simplify the existing game-playing or Maurer’s-methodology-based proofs. For example, see the proof of ENR, Fesitel ciphers, and SoP. However, in some cases, we unify the proofs of various related schemes into one general result. For example, see the general proof of ${\mathrm{NR}}^{⋆}$. For SPRP enciphering schemes, we provide the first proof in the H-technique.

#### Organization of the Paper

We begin by developing the notations and conventions, in Section 2, that will be used in the paper. In Section 3, we formalize the model for bounded-query interactive algorithms. In Section 4, we describe the H-technique tool and its variant, the expectation method. We also present a brief on how to capture the random system methodology using the H-technique. In Section 5, Section 6 and Section 7, we give alternate proofs for some hash-based schemes, Feistel-like schemes, and popular SPRP schemes, respectively. Section 8 gives proofs for beyond-birthday-bound secure PRFs, namely, SoP and SoEM22. In Section 9, we prove the optimality of the H-technique and use similar ideas to present an alternate proof for the non-adaptive to adaptive PRP composition.

## 2. Preliminaries

### 2.1. Notation

We simply write the set {1,2,…,m} as [m]. We denote a *q*-tuple $\left(x_{1},\ldots ,x_{q}\right)$ as $x^{q}$. We sometime use the notation $\left\{x^{q}\right\}$ to denote the set $\left\{x_{i}:1\leq i\leq q\right\}$. For a *q*-tuple $v=v^{q}$, we sometimes denote $v_{i}$ as ${v|}_{i}$. A binary sequence $b^{q}$ is called monotone if $b_{i+1}\geq b_{i}$ for all i∈[q−1]. Thus, any binary monotone sequence must be of the form $0^{i}1^{q-i}$ for some *i*.

For a set X, we write $X^{\left(r\right)}$ for the set of all *r* tuples $x^{r}\in X^{r}$ such that $x_{1},\ldots ,x_{r}$ are distinct. We write N(N−1)⋯(N−r+1) as ${\left(N\right)}_{r}$. If the size of the set X is *N*, then clearly, $|X^{\left(r\right)}{|=\left(N\right)}_{r}$.

A function g(a,b) is functionally independent of *b* if for all $a,b,b^{\prime }$, $g\left(a,b\right)=g\left(a,b^{\prime }\right)$. In this case there exists a function $g^{\prime }$ such that for all a,b, $g\left(a,b\right)=g^{\prime }\left(a\right)$.

Given an index set I, we denote an indexed family (or a tuple) as ${\left\{x_{i}\right\}}_{i\in I}$ or $x^{I}$. Note that it is different from the set $\left\{x_{i}\;:\;i\in I\right\}$. For some i≠j, $x_{i}$ and $x_{j}$ may be the same and we ignore repetition in the set representation, whereas in the indexed family we allow repetition. More formally, it can be represented as a function from the index set to some set in which $x_{i}$ values belong. Note that $x^{q}$ is a shorthand notation for $x^{\left[q\right]}$, where [q] is the index set.

Notations on Compatibility. The set of all functions from X to Y is denoted as Func(X,Y). Similarly, the set of all permutations over Y is denoted as Perm(Y).

A pair of tuples $\left(x^{q},y^{q}\right)$ is referred to as function-compatible if $x_{i}=x_{j}\Rightarrow y_{i}=y_{j}$. We denote it as $x^{q}⇝y^{q}$.A pair of tuples $\left(x^{q},y^{q}\right)$ is referred to as permutation-compatible if $x_{i}=x_{j}\Leftrightarrow y_{i}=y_{j}$. We denote it as $x^{q}↭y^{q}$.A pair of triples $\left(t^{q},x^{q},y^{q}\right)$ is referred to as tweakable-permutation-compatible if $\left(t_{i},x_{i}\right)=\left(t_{j},x_{j}\right)\Leftrightarrow \left(t_{i},y_{i}\right)=\left(t_{j},y_{j}\right)$. We denote it as $x^{q}\overset{t^{q}}{↭}y^{q}$ (equivalently, $\left(t^{q},x^{q}\right)↭\left(t^{q},y^{q}\right)$).

### 2.2. Statistical Distance

Statistical distance (also known as total variation [100] in the statistics community) is a metric on the set of probability functions over a finite set Ω. This is the most common metric in cryptography. As we see later, it has a close relationship with the distinguishing advantage.

**Definition** **1**(statistical distance)**.**
*Let ${\mathrm{Pr}}_{0}$ and ${\mathrm{Pr}}_{1}$ be two probability functions over a finite set *Ω*. We define the statistical distance between ${\mathrm{Pr}}_{0}$ and ${\mathrm{Pr}}_{1}$ as*
$$∥{\mathrm{Pr}}_{0}-{\mathrm{Pr}}_{1}∥\;:=\;\frac{1}{2}\sum_{x\in \Omega }\left|{\mathrm{Pr}}_{0}\left(x\right)-{\mathrm{Pr}}_{1}\left(x\right)\right|.$$
*When X,Y are two random variables over *Ω*, we define $\Delta \left(X\;;\;Y\right)=∥{\mathrm{Pr}}_{X}-{\mathrm{Pr}}_{Y}∥$.*

It is easy to verify that the statistical distance satisfies the symmetry and **triangle inequality**. Moreover, it always lies between zero and one. It is one if and only if the support (the set of all elements of having positive probabilities) of the probability distributions are disjoint and it is zero if and only if the distributions are the same.

The following properties indicate that the total variation is a bounded metric over the set of all probability functions.

**Lemma** **1**(Properties)**.**
*For any probability functions*
${\mathrm{Pr}}_{1},{\mathrm{Pr}}_{2},\ldots ,{\mathrm{Pr}}_{d}$*, we have*
*1.* *(non-negative) $∥{\mathrm{Pr}}_{1}-{\mathrm{Pr}}_{2}∥\geq 0$.**2.* *(identification) $∥{\mathrm{Pr}}_{1}-{\mathrm{Pr}}_{2}∥=0$ if and only if ${\mathrm{Pr}}_{1}={\mathrm{Pr}}_{2}$.**3.* *(symmetric) $∥{\mathrm{Pr}}_{1}-{\mathrm{Pr}}_{2}∥=∥{\mathrm{Pr}}_{2}-{\mathrm{Pr}}_{1}∥$.**4.* **Triangle inequality***(general form):*$$∥{\mathrm{Pr}}_{1}-{\mathrm{Pr}}_{d}∥\leq ∥{\mathrm{Pr}}_{1}-{\mathrm{Pr}}_{2}∥+\cdots +∥{\mathrm{Pr}}_{d-1}-{\mathrm{Pr}}_{d}∥.$$*5.* *$∥{\mathrm{Pr}}_{1}-{\mathrm{Pr}}_{2}∥\leq 1$. The equality holds if and only if the supports of these two probability functions are disjoint.*

All these results are easy to verify based on the definition of statistical distance. We leave this as an exercise for the reader.

**Definition** **2.** 
*Given two probability distributions*

${\mathrm{Pr}}_{0}$

*and*

${\mathrm{Pr}}_{1}$

*over *Ω*, we associate two sets*

$$\begin{array}{cc} \Omega_{>} & =\{x\in \Omega :{\mathrm{Pr}}_{0}\left(x\right)>{\mathrm{Pr}}_{1}\left(x\right)\},and\;\\ \Omega_{\geq } & =\{x\in \Omega :{\mathrm{Pr}}_{0}\left(x\right)\geq {\mathrm{Pr}}_{1}\left(x\right)\}. \end{array}$$



**Lemma** **2.** 
*For any two probability distributions*

${\mathrm{Pr}}_{0}$

*and*

${\mathrm{Pr}}_{1}$

*, we have*

$$\max_{E}\left({\mathrm{Pr}}_{0}\left(E\right)-{\mathrm{Pr}}_{1}\left(E\right)\right)=\sum_{x\in \Omega }\max\left\{0,{\mathrm{Pr}}_{0}\left(x\right)-{\mathrm{Pr}}_{1}\left(x\right)\right\}\;=\;∥{\mathrm{Pr}}_{0}-{\mathrm{Pr}}_{1}∥.$$

*The maximum is achieved at*

E

*if and only if*

$\Omega_{>}\subseteq E\subseteq \Omega_{\geq }$

*.*


**Proof.** It is easy to see that the maximum value of ${\mathrm{Pr}}_{0}\left(E\right)-{\mathrm{Pr}}_{1}\left(E\right)$ is achieved at E if and only if $\Omega_{>}\subseteq E\subseteq \Omega_{\geq }$ (for any $x\notin \Omega_{\geq }$, the contribution ${\mathrm{Pr}}_{0}\left(x\right)-{\mathrm{Pr}}_{1}\left(x\right)$ is negative). Here, we can note that
$$\max\left\{0,{\mathrm{Pr}}_{0}\left(x\right)-{\mathrm{Pr}}_{1}\left(x\right)\right\}=\left\{\begin{array}{cc} 0 & \;\mathrm{if}\;x\notin \Omega_{>}\\ {\mathrm{Pr}}_{0}\left(x\right)-{\mathrm{Pr}}_{1}\left(x\right) & \;\mathrm{if}\;x\in \Omega_{>}. \end{array}\right.$$
So,
$$\begin{array}{cc} \sum_{x\in \Omega }\max\left\{0,{\mathrm{Pr}}_{0}\left(x\right)-{\mathrm{Pr}}_{1}\left(x\right)\right\} & =\sum_{x\in \Omega_{>}}{\mathrm{Pr}}_{0}\left(x\right)-{\mathrm{Pr}}_{1}\left(x\right)\\ \; & =\max_{E}\left({\mathrm{Pr}}_{0}\left(E\right)-{\mathrm{Pr}}_{1}\left(E\right)\right). \end{array}$$
This proves the first equality. Now, we write
$$2∥{\mathrm{Pr}}_{0}-{\mathrm{Pr}}_{1}∥=\sum_{x\in \Omega_{>}}|{\mathrm{Pr}}_{0}\left(x\right)-{\mathrm{Pr}}_{1}\left(x\right)|+\sum_{x\notin \Omega_{>}}\left|{\mathrm{Pr}}_{0}\left(x\right)-{\mathrm{Pr}}_{1}\left(x\right)\right|.$$
The first sum can be simplified as
$$\sum_{x\in \Omega_{>}}{\mathrm{Pr}}_{0}\left(x\right)-{\mathrm{Pr}}_{1}\left(x\right)={\mathrm{Pr}}_{0}\left(\Omega_{>}\right)-{\mathrm{Pr}}_{1}\left(\Omega_{>}\right).$$
Similarly, the second sum can be simplified to
$$\begin{array}{cc} \sum_{x\notin \Omega_{>}}{\mathrm{Pr}}_{1}\left(x\right)-{\mathrm{Pr}}_{0}\left(x\right) & ={\mathrm{Pr}}_{1}\left(\Omega_{>}^{c}\right)-{\mathrm{Pr}}_{0}\left(\Omega_{>}^{c}\right)\\ \; & ={\mathrm{Pr}}_{0}\left(\Omega_{>}\right)-{\mathrm{Pr}}_{1}\left(\Omega_{>}\right) \end{array}$$
If we add these two sums, we obtain the second equality. □

**Corollary** **1.** 
*Let*

$X_{0}∼{\mathrm{Pr}}_{0}$

*and*

$X_{1}∼{\mathrm{Pr}}_{1}$

*. Let*

$\epsilon_{\mathrm{opt}}\left(x\right)=\max\left\{0,1-\frac{{\mathrm{Pr}}_{1}\left(x\right)}{{\mathrm{Pr}}_{0}\left(x\right)}\right\}$

*for all x in the support of*

$X_{0}$

*. Then,*

$$∥{\mathrm{Pr}}_{0}-{\mathrm{Pr}}_{1}∥\;=\;\mathrm{Ex}\left[\epsilon_{\mathrm{opt}}\left(X_{0}\right)\right].$$



## 3. Models for Interactive Algorithms

### 3.1. Probabilistic Function

A probabilistic function (defined below) is a mathematical model for the black-box behavior of a probabilistic algorithm. We also use the same object to model probabilistic interactive algorithms.

**Definition** **3**(probabilistic function)**.**
*A* probabilistic function *with an input space*
X
*and an output space*
Y
*is a function*
f:R×X→Y
*for some finite set*
R*, called the*
**random coin space***. We can also simply write (abusing notation)*
$f:X\overset{*}{\to }Y$
*suppressing the notation for the random coin space.*

If the random coin space is a singleton (i.e., degenerated) we simply ignore the random coin space. In this case, the probabilistic function is reduced to a function. Given an input x∈X, we first sample $R\overset{*}{\leftarrow }R$ (in most cases uniformly) and then we define an output random variable f(x):=f(R,x) over Y. So, for all y∈Y,
(1)$$p_{x}^{f}\left(y\right):=\mathrm{Pr}\left[f\left(x\right)=y\right]=\sum_{\begin{array}{c} r\\ f(r,x)=y \end{array}}\mathrm{Pr}\left[R=r\right]$$

**Definition** **4.** 
*With each probabilistic function*

$f:X\overset{*}{\to }Y$

*, we associate a family of probability functions over*

Y

*(indexed by the input space*

X

*)*

$$p^{f}:=\left\{p_{x}^{f}\;|\;x\in X\right\},$$

*where*

$p_{x}^{f}\left(y\right):=\mathrm{Pr}\left[f\left(x\right)=y\right]$

*. We call the*

$p^{f}$

**the probabilistic system**
*associated with the probabilistic function f.*


Note that the probabilistic function and probabilistic system are analogous to a random variable and its probability distribution.

**Example** **1**(Keyed Functions)**.**
*This is an important example for cryptography. Many cryptographic designs are viewed as keyed functions. Let F be a keyed function family*
$\left\{F_{k}\;|\;k\in K\right\}$
*such that for all keys*
k∈K*,*
$F_{k}:X\to Y$*.*
*We sample key*

$K\leftarrow_{\$}K$

*and treat it as a random coin, and we obtain a probabilistic function (abusing notation)*

$F:X\overset{*}{\to }Y$

*, mapping x to*

$F\left(K,x\right):=F_{K}\left(x\right)$

*(also written as*

K(x)

*, whenever*

K

*actually represents the function F).*


**Notation** **1.** 
*Given a probabilistic function*

$f:X\overset{*}{\to }Y_{1}\times Y_{2}$

*we write*

$f=(f_{1},f_{2})$

*where*

$f\left(r,x\right)=(f_{1}\left(r,x\right),f_{2}\left(r,x\right))$

*and*

$f_{i}:X\overset{*}{\to }Y_{i}$

*,*

i=1,2

*. The probabilistic functions*

$f_{1}$

*and*

$f_{2}$

*are basically two components of f and we also call them truncated probabilistic functions. This can be similarly extended for the Cartesian product of more than two sets.*


### 3.2. Function Models of Interactive Algorithms and Their Interaction

An interactive algorithm is modeled as a (probabilistic) interactive Turing machine [1,101]. In this paper, probabilistic functions are modeled for interactive algorithms. This model is general enough to capture finite and bounded interactions between two interactive algorithms (i.e., the number of interactions between two algorithms is bounded by some fixed positive integer, say *q*).

**Definition** **5**(function models of interactive algorithms)**.**
*Let q be a positive integer.*
*Joint Response Function**:- A q*-joint (X,Y) response function *is a probabilistic function $F:X^{q}\overset{*}{\to }Y^{q}$ such that for all random coins r, the mapping $x^{q}\mapsto F\left(r,x^{q}\right){|}_{i}$ is* functionally independent of $x_{i+1},\ldots ,x_{q}$.*Joint Query Function**:- A probabilistic function $A:Y^{q}\overset{*}{\to }X^{q}$ is called a**q*-joint (X,Y) query function *if for all random coins r, the mapping $y^{q}\mapsto A\left(r,y^{q}\right){|}_{i}$ is*functionally independent of $y_{i},\ldots ,y_{q}$*. Moreover, it is called*
-**nonadaptive** 
*if $A(r,y^{q})$ is functionally independent of $y^{q}$ and*-**deterministic** 
*if the random coin space is a singleton (we simply drop the random coin space notation and write it as a function $A:Y^{q}\to X^{q}$).*

We can also simply refer to a *q*-joint (X,Y) query function and a *q*-joint (X,Y) response function as a (X,Y) joint query function and a (X,Y) joint response function respectively.

The joint query and response function together capture the interaction. A joint query function captures the functional view of an interactive algorithm that initiates the interaction and a joint response function captures the functional view of the corresponding oracle algorithm. When a joint query function A interacts with a joint response function F, $x_{1}$ only depends on the random coin of A, whereas $y_{1}$ depends on $x_{1}$ and the random coin of F. Similarly, $x_{2}$ depends on $y_{1}$ and its random coin, and $y_{2}$ depends on $x_{1},x_{2}$ and its random coin. In this way, we can define $x^{q}$ and $y^{q}$ based on random coins of A and F. The pair $\left(x^{q},y^{q}\right)$ is called a transcript (which is a function of the pairs of random coins of A and F).

We now formally define the transcript random variable. Based on the given conditions of the definitions of the joint response and query functions, there exist functions $A_{i}$ and $F_{i}$, i∈[q], such that for all $y^{q}$, $A\left(r,y^{q}\right){|}_{i}=A_{i}\left(r,y^{i-1}\right)$ and for all $x^{q}$, $F\left(r^{\prime },x^{q}\right){|}_{i}=F_{i}\left(r^{\prime },x^{i}\right)$.

**Definition** **6**(transcript)**.**
*Let*
A
*and*
F
*be the*
(X,Y)
*joint query function and joint response function, respectively. Let*
$A_{i}$
*and*
$F_{i}$
*be defined as above. We define the* transcript *random variable as*
$\tau \left(A^{F}\right)=\left(X^{q},Y^{q}\right)$*, where*
$X_{i}$
*values and*
$Y_{i}$
*values are defined recursively as follows:*
$$X_{i}=A_{i}\left(R,Y^{i-1}\right),\;\;Y_{i}=F_{i}\left(R^{\prime },X^{i}\right),\;\;1\leq i\leq q$$
*and*
R
*and*
$R^{\prime }$
*are random coins of*
A
*and*
F*, respectively.*

Based on the above definition, it is clear that for any fixed random coins *r* and $r^{\prime }$, the transcript is the unique pair $\left(x^{q},y^{q}\right)$ such that $A\left(r,y^{q}\right)=x^{q}$ and $F\left(r^{\prime },x^{q}\right)=y^{q}$. So for any $\left(x^{q},y^{q}\right)\in X^{q}\times Y^{q}$, using the independence of the random coins A and F, we have
(2)$$\begin{array}{cc} \mathrm{Pr}[\tau \left(A^{F}\right)=\left(x^{q},y^{q}\right)] & =\mathrm{Pr}\left[A\left(y^{q}\right)=x^{q}\right]\times \mathrm{Pr}\left[F\left(x^{q}\right)=y^{q}\right]. \end{array}$$
In terms of the probabilistic systems $p^{A}$ and $p^{F}$ associated with A and F, respectively (see Definition 4), we can write the probability, realizing a transcript $\tau =(x^{q},y^{q})$ as
$$\begin{array}{cc} \mathrm{Pr}[\tau \left(A^{F}\right)=\left(x^{q},y^{q}\right)] & =p_{y^{q}}^{A}\left(x^{q}\right)\times p_{x^{q}}^{F}\left(y^{q}\right) \end{array}$$
So, the transcript probability is determined by the probabilistic systems $p^{A}$ and $p^{F}$.

Extended Transcript. The transcript is a piece of information obtained by the joint query function through an interaction. Sometimes we release an extra piece of information, say S, in addition to the transcript, to the adversary. This is given only after all interaction is completed.In other words, the queries $x^{q}$ cannot functionally depend on S, whereas S can depend on queries. To formalize this, let us define an extended response function.

**Definition** **7**(extended transcript)**.** *An*
S*-extended*
(X,Y) joint response function *is a probabilistic function*
$\bar{F}=\left(F,S\right):X^{q}\overset{*}{\to }Y^{q}\times S$*. For any*
(X,Y)
*joint query function*
A*, we define the* (extended) transcript of $A^{\bar{F}}$
*as*
$$\bar{\tau }\left(A^{F}\right)=\tau \left(A^{\bar{F}}\right)=\left(\tau \left(A^{F}\right)\;,\;S\left(X^{q}\right)\right)\;:=\;\left(\tau \left(A^{F(R,\cdot )}\right)\;,\;S\left(R,X^{q}\right)\right),$$
*where*
R
*denotes the random coin of*
$\bar{F}$
*and*
$\tau \left(A^{F}\right)=\left(X^{q},Y^{q}\right)$*. We call*
S the adjoined random variable *to*
F*.*

MBO Extension. Now we describe a popular extended joint responsefunction. An MBO (monotone binary output) extension $\bar{F}$ is an ${\left\{0,1\right\}}^{q}$-extension of a joint response function F such that the support of the adjoined random variable S is the set of all monotone binary sequences. We call the extended transcript $\tau (A^{\bar{F}})$ good if $S=0^{q}$, otherwise we call it bad. Informally, S denotes whether some bad event occurred upon some query or not. This is accomplished by setting the bit corresponding to the query index to 1. Since S is monotone in nature, whenever the bad flag is set to 1, it continues to be 1 for the rest of the queries. This justifies the fact that the support of S is $\left\{0^{i}1^{q-i}:0\leq i\leq q\right\}$.

Later we will see that a simpler and equally powerful extension would be to release a binary variable B to denote whether a bad event happened or not in the whole transcript. Thus, B=0 if $S=0^{q}$; otherwise, B=1. We adjoin the random variable B only, instead of an MBO S.

For an extended system $\bar{F}=\left(F,S\right)$, we can similarly associate a probabilistic system, defined as
(3)$$p_{x^{q}}^{\bar{F}}\left(y^{q},s\right)=\mathrm{Pr}\left[F\left(x^{q}\right)=y^{q},S=s\right]$$
For any $\left(x^{q},y^{q},s\right)\in X^{q}\times Y^{q}\times S$, we have
$$\mathrm{Pr}\left[\tau \left(A^{\bar{F}}\right)=\left(x^{q},y^{q},s\right)\right]=\mathrm{Pr}\left[A\left(y^{q}\right)=x^{q}\right]\times \mathrm{Pr}\left[F\left(x^{q}\right)=y^{q},S=s\right].$$

### 3.3. Examples of Response Functions

**Keyed Function**. Let *F* be a keyed function family $\left\{F_{k}\;|\;k\in K\right\}$ such that for all keys k∈K, $F_{k}:X\to Y$. We can also view *F* as a function F:K×X→Y, where $F\left(k,x\right)=F_{k}\left(x\right)$. If we choose the key $K\leftarrow_{\$}K$ and treat it as a random coin, we obtain a joint response function (we call it a (deterministic) keyed function) as
$$F\left(k,x^{q}\right)=\left(F\left(k,x_{1}\right),\ldots ,F\left(k,x_{q}\right)\right),\;\;x^{q}\in X^{q}.$$
**Keyed Strong Permutation**. When F(k,·) is a permutation on Y for all keys k∈K, one can consider an interaction in which a joint query function makes queries to the inverse function as well. To capture this, we associate a new keyed function
$$F_{k}^{\pm }:\left\{1,-1\right\}\times Y\to Y$$
mapping (1,x) to $F_{k}\left(x\right)$ and mapping (−1,x) to $F_{k}^{-1}\left(x\right)$. We also write $F_{k}\left(\delta ,x\right):=F_{k}^{\delta }\left(x\right)$. The joint response function associated to the keyed function $F^{\pm }$ is denoted as $F^{\pm }$ and we call it keyed strong permutation.

**Definition** **8.**
*Given a triple of tuples*

$\left(\delta^{q},x^{q},y^{q}\right)\in {\left\{1,-1\right\}}^{q}\times Y^{2q}$

*we associate a*
**forward-only representation**

$\left(\delta^{q},a^{q},b^{q}\right)$

*as*

$$\left(a_{i},b_{i}\right)=\left\{\begin{array}{cc} \left(x_{i},y_{i}\right) & if\;\delta_{i}=1\\ \left(y_{i},x_{i}\right) & otherwise. \end{array}\right.$$

*The forward-only representation is an equivalent representation of the original triple, as we can uniquely reconstruct the original triple from it.*


Suppose that $\left(\delta^{q},a^{q},b^{q}\right)$ is a *forward-only representation* of $\left(\delta^{q},x^{q},y^{q}\right)$. Then, $F_{k}\left(a_{i}\right)=b_{i}$ for all *i* if and only if $F_{k}^{\delta_{i}}\left(x_{i}\right)=y_{i}$ for all *i*. So for every $\left(\delta^{q},x^{q},y^{q}\right)$, we have
$$\mathrm{Pr}\left[F^{\pm }\left(\delta^{q},x^{q}\right)=y^{q}\right]=\mathrm{Pr}\left[F\left(a^{q}\right)=b^{q}\right].$$
So, *the probabilistic system associated to $F^{\pm }$ is completely determined by the probabilistic system associated with F.*

#### Some Ideal Random Systems

We describe some popular ideal random systems. Let X, Y and T be finite sets such that N=|Y|.

**Definition** **9**(random function)**.** *A*
(X,Y)
*random function is an*
(X,Y)
*joint response function ρ such that for all*
$x^{q}\in X^{q}$
*and*
$y^{q}\in Y^{q}$
*with*
$x^{q}⇝y^{q}$
*(function compatible),*
$$\mathrm{Pr}\left[\rho \left(x^{q}\right)=y^{q}\right]=N^{-s}$$
*where s is the number of distinct x values present in*
$x^{q}$*. In all other cases the probability is zero.*

**Definition** **10**(random permutation)**.** *A*
Y
*random permutation is an*
(Y,Y)
*joint response function ***π*** such that for all*
$x^{q},y^{q}\in Y^{q}$
*with*
$x^{q}↭y^{q}$
*(permutation compatible),*
$$\mathrm{Pr}\left[\pi \left(x^{q}\right)=y^{q}\right]=\frac{1}{{\left(N\right)}_{s}}$$
*where s is the number of distinct*
$x_{i}$
*values present in*
$x^{q}$*. In all other cases the probability is zero.*

As described before, we can also similarly define a strong random permutation $\pi^{\pm }$ which provides the access of inverse. More precisely, for any $\left(\delta^{q},x^{q},y^{q}\right)$, $\mathrm{Pr}\left[\pi^{\pm }\left(\delta^{q},x^{q}\right)=y^{q}\right]=\frac{1}{{\left(N\right)}_{s}}$, provided that $a^{q}↭b^{q}$, where $\left(\delta^{q},a^{q},b^{q}\right)$ is the forward-only transcript of $\left(\delta^{q},x^{q},y^{q}\right)$ and *s* is the number of distinct $a_{i}$ values present in $a^{q}$ (which is same as the number of distinct values present in $b^{q}$).

We have defined the above ideal systems through their probabilistic systems. One can define these through deterministic keyed functions. For a random function, the key space is Func(X,Y), the set of all functions from X to Y. For any k∈Func(X,Y), and x∈X, we define ρ(k,x)=k(x). For a random permutation, the key space is Perm(Y), the set of all permutations over Y. For any k∈Perm(Y), and x∈Y, we define π(k,x)=k(x).

One can easily verify that the probabilistic systems view is same as the deterministic keyed function view. The two views are actually the same functions defined over two different domains.

Tweakable Random Permutation. Given a tweakable-permutation-compatible tuple $\left(t^{q},x^{q},y^{q}\right)$, we associate a tuple of positive numbers $\left(c_{1},\ldots ,c_{r}\right)$ as follows. Let $t_{1}^{\prime },\ldots ,t_{r}^{\prime }$ denote the distinct tweaks present in $t^{q}$. We write $\mathrm{mcoll}\left(t^{q}\right)=c^{r}$ where $c_{i}=\left|\left\{x_{j}:t_{j}=t_{i}^{\prime }\right\}\right|$. Clearly, $\sum_{i}c_{i}=q$ when $\left(t^{q},x^{q},y^{q}\right)$ is a tweakable-permutation-compatible tuple. Basically, $\sum_{i}c_{i}$ represents the number of distinct $\left(t_{i},x_{i}\right)$ pairs present in $\left(t^{q},x^{q}\right)$.

**Definition** **11**(tweakable random permutation)**.**
*A*
(T,Y)
*tweakable random permutation is a*
(T×Y,Y)
*joint response function*
$\tilde{\pi }$
*such that for all tweakable-permutation-compatible tuples*
$\left(t^{q},x^{q},y^{q}\right)$
*with*
$\mathrm{mcoll}\left(t^{q}\right)=c^{r}$*,*
(4)$$\mathrm{Pr}\left[\tilde{\pi }\left(t^{q},x^{q}\right)=y^{q}\right]=\prod_{i=1}^{r}\frac{1}{\;{\left(N\right)}_{c_{i}}}.$$
*The probability is zero for all other tuples*
$\left(t^{q},x^{q},y^{q}\right)$*.*

We can write the above probability in another equivalent form. For each *i*, we define $s_{i}$ as the number of j<i, such that $t_{j}=t_{i}$. Then, we have
(5)$$\mathrm{Pr}\left[\tilde{\pi }\left(t^{q},x^{q}\right)=y^{q}\right]=\prod_{i=1}^{q}\frac{1}{N-s_{i}}.$$
Intuitively, when we respond to the *i*th query $\left(t_{i},x_{i}\right)$, we look at all those *j* for which $t_{j}=t_{i}$. Let $S_{i}$ be the set of all $y_{j}$ values for which $t_{j}=t_{i}$. The response of the *i*th query is to select an element randomly from $S_{i}^{c}$ (in other words, without a replacement sample for the same tweak values).

To realize this probabilistic system, we define a keyed function corresponding to it. Let the key space be Func(T,Perm(Y)), the set of all functions from the tweak space to the set of all permutations. Thus, if *k* is a key and *t* is a tweak, k(t) is a permutation over Y. We write k(t)(x) as k(t,x) or $\tilde{\pi }\left(k,\left(t,x\right)\right)$. One can again check that the probabilistic system associated with this joint response function is the same as the tweakable random permutation as defined above.

## 4. H-Technique Tools

### 4.1. Distinguisher and Its Advantage

Let F and G be two (X,Y) joint response functions and A be a (X,Y) joint query system with random coin space R. Let $b:R\times X^{q}\times Y^{q}\to \left\{0,1\right\}$ be a binary function (also called a decision function). We call the pair (A,b), denoted as $A_{b}$, a **distinguisher**.

The algorithm A obtains a transcript $\tau =(x^{q},y^{q})$.The function *b* finally makes a decision based on the transcript and the random coin initially sampled by A.

More formally, the output of $A_{b}^{F}$ is $b(R,\tau \left(A^{F}\right))$ where R is the random coin of A, which is used to generate the transcript $\tau (A^{F})$. We now define


$\Delta_{A_{b}}\left(F\;;\;G\right)\;:=\;|\mathrm{Pr}\left[A_{b}^{F}\to 1\right]-\mathrm{Pr}\left[A_{b}^{G}\to 1\right]|$.


Let E be the set of all tuples $\left(r,x^{q},y^{q}\right)$ for which *b* returns 1. From the equivalent definition of statistical distance (see Lemma 2) we have
(6)$$\begin{array}{cc} \Delta_{A_{b}}\left(F\;;\;G\right) & \leq \Delta (\left(R,\tau \left(A^{F}\right)\right)\;;\;\left(R,\tau \left(A^{G}\right)\right)). \end{array}$$
Moreover, equality is achieved if we define the decision function, called the optimal decision function and denoted $b_{opt}$, as follows:
$$b_{opt}\left(r,x^{q},y^{q}\right)=1\Leftrightarrow \mathrm{Pr}\left[R=r,\tau \left(A^{F}\right)=\left(x^{q},y^{q}\right)\right]\geq \mathrm{Pr}\left[R=r,\tau \left(A^{G}\right)=\left(x^{q},y^{q}\right)\right].$$
Complexity. Note that the computation of $b_{opt}$ may not be efficient. In general, we consider two types of complexities for an adversary (both for the query system and the decision function) to measure the efficiency of an algorithm. One type considers all computational complexities, which includes, e.g., time, memory, etc. The other type considers the data complexities, which includes the number of queries (which is *q* in our case), the total number of bits in all queries, the size of the largest queries, etc. As we are interested in information-theoretic analysis, we only keep complexity related to oracle calls and we consider only **unbounded time adversaries**. We always assume that the decision-making function *b* is optimum and hence, $\Delta_{A_{b}}\left(F\;;\;G\right)=\Delta \left(\left(R,\tau \left(A^{F}\right)\right)\;;\;\left(R,\tau \left(A^{G}\right)\right)\right)$. Thus, we simply denote a distinguisher as A (by its joint query function), ignoring the notation *b*.

#### Conventions

Now, we state some conventions which can be assumed without the loss of generality in this paper. This will simplify the process of analyzing the distinguishing advantage.
**Distinguishers are deterministic**: Given any query function A, and for a fixed random coin *r*, let A[r]:=A(r,·) denote the deterministic query function which basically runs A with the random coin *r*. It is easy to verify that $\Delta_{A}\left(F\;;\;G\right)={\mathrm{Ex}}_{R}\left(\Delta_{A[R]}\left(F\;;\;G\right)\right)$ and hence there exists $r_{0}$ for which ${\mathrm{Ex}}_{R}\left(\Delta_{A[R]}\left(F\;;\;G\right)\right)\leq \Delta_{A[r_{0}]}\left(F\;;\;G\right)$. Hence,
$$\Delta_{A}\left(F\;;\;G\right)\leq \Delta_{A[r_{0}]}\left(F\;;\;G\right).$$**No redundant queries**: In this paper we only consider deterministic keyed functions. An adversary A interacting with a deterministic keyed function is called redundant if A makes two identical queries (i.e., $x_{i}=x_{j}$ for some i<j). An adversary A interacting with a deterministic keyed strong permutation $F^{\pm }$ is called redundant if for some i<j, $\left(\delta_{j},x_{j}\right)=\left(\delta_{i},x_{i}\right)$ or $\left(\delta_{j},x_{j}\right)=\left(-\delta_{i},y_{i}\right)$, where $y_{i}$ is the response of the *i*th query. Note that, in this case, $\left(a_{j},b_{j}\right)=\left(a_{i},b_{i}\right)$, where $\left(\delta^{q},a^{q},b^{q}\right)$ denotes the forward-only transcript. The response of *j*th query is uniquely determined from the *i*th query. Similarly, we define redundant queries for a tweakable keyed permutation. The *i*th query $\left(\delta_{i},t_{i},x_{i}\right)$ is called redundant if there is j<i with $t_{j}=t_{i}$, either $\left(\delta_{j},x_{j}\right)=\left(\delta_{i},x_{i}\right)$ or $\left(\delta_{j},y_{j}\right)=\left(-\delta_{i},x_{i}\right)$.

Note that for all redundant queries, the response is uniquely determined from the previous query-responses and hence without the loss of generality we may ignore those queries. Thus, *we assume that all such adversaries are non-redundant*.

### 4.2. Security Definitions

Here we define PRF, PRP, SPRP, and their tweakable versions against adaptive and nonadaptive adversaries. Let $A(\theta_{D})$ (and $A_{\mathrm{na}}\left(\theta_{D}\right)$) denote the set of all adversaries A, using at most $\theta_{D}$ data complexity in adaptive ways (and nonadaptive ways, respectively). If the computational complexity is unbounded (or infinity) in all these definitions, we simply drop the notation $\theta_{C}$.



${\mathrm{Adv}}_{F}^{\mathrm{prf}}\left(\theta_{D}\right)=\max_{A\in A(\theta_{D})}\Delta_{A}\left(F\;;\;\rho \right)$,${\mathrm{Adv}}_{F}^{\mathrm{nprf}}\left(\theta_{D}\right)=\max_{A\in A_{\mathrm{na}}\left(\theta_{D}\right)}\Delta_{A}\left(F\;;\;\rho \right)$,

${\mathrm{Adv}}_{F}^{\mathrm{prp}}\left(\theta_{D}\right)=\max_{A\in A(\theta_{D})}\Delta_{A}\left(F\;;\;\pi \right)$,${\mathrm{Adv}}_{F}^{\mathrm{sprp}}\left(\theta_{D}\right)=\max_{A\in A(\theta_{D})}\Delta_{A}\left(F^{\pm }\;;\;\pi^{\pm }\right)$,${\mathrm{Adv}}_{F}^{\mathrm{nprp}}\left(\theta_{D}\right)=\max_{A\in A_{\mathrm{na}}\left(\theta_{D}\right)}\Delta_{A}\left(F\;;\;\pi \right)$,

${\mathrm{Adv}}_{F}^{\mathrm{tprp}}\left(\theta_{D}\right)=\max_{A\in A(\theta_{D})}\Delta_{A}\left(F\;;\;\tilde{\pi }\right)$,${\mathrm{Adv}}_{F}^{\mathrm{tsprp}}\left(\theta_{D}\right)=\max_{A\in A(\theta_{D})}\Delta_{A}\left(F^{\pm }\;;\;{\tilde{\pi }}^{\pm }\right)$.



### 4.3. H-Technique

Here, we describe the extended version of the H-technique. The basic or standard version, also called the Coefficients H technique, is a simple instantiation of the extended version (viewing the adjoined random variable as a degenerated or fixed constant).

**Lemma** **3**(Extended H-technique)**.** *Suppose that*
$\bar{F}:=\left(F,S\right)$
*and*
G¯:=(G,S′)
*are two*
S*-extended*
(X,Y)
*response systems. Let *Ω* denote the set of all attainable transcripts, i.e., the support of*
${\mathrm{Pr}}_{\bar{G}}$*. Suppose that there is a set*
$\Omega_{\mathrm{bad}}\subseteq \Omega$
*such that for all*
$\left(x^{q},y^{q},s\right)\notin \Omega_{\mathrm{bad}}$*,*
$$\begin{array}{cc} \frac{\mathrm{Pr}[F\left(x^{q}\right)=y^{q},S=s]}{\mathrm{Pr}[G\left(x^{q}\right)=y^{q},S^{\prime }=s]} & \geq (1-\epsilon ) \end{array}$$
*for some*
ϵ≥0*. Then, for any*
(X,Y)
*adversary*
A*,*
(7)$$\begin{array}{cc} \Delta_{A}\left(F;G\right)\leq \Delta \left(\bar{\tau }\left(A^{F}\right);\bar{\tau }\left(A^{G}\right)\right) & \leq \mathrm{Pr}[\left(\tau \left(A^{G}\right),S^{\prime }\right)\in \Omega_{\mathrm{bad}}]+\epsilon . \end{array}$$

A proof of the H-technique is presented, among others, in [36,38,44,45]. Here, we provide a short proof for the sake of completeness.

**Proof.** For any adversary A, it is easy to see that $\Delta_{A}\left(F;G\right)\leq \Delta \left(\bar{\tau }\left(A^{F}\right);\bar{\tau }\left(A^{G}\right)\right)$. This holds as the decision-making function is free to discard the additional information. Let ${\mathrm{Pr}}_{0}={\mathrm{Pr}}_{\bar{G}}$ and ${\mathrm{Pr}}_{1}={\mathrm{Pr}}_{\bar{F}}$. Then, $\Omega_{>}$ is well-defined (see Definition 2). According to Lemma 2, we have
$$\begin{array}{cc} \Delta (\bar{\tau }\left(A^{F}\right);\bar{\tau }\left(A^{G}\right)) & =\sum_{\omega \in \Omega }\max\left\{0,{\mathrm{Pr}}_{\bar{G}}\left(\omega \right)-{\mathrm{Pr}}_{\bar{F}}\left(\omega \right)\right\}\\ \; & =\sum_{\omega \in \Omega_{>}}{\mathrm{Pr}}_{\bar{G}}\left(\omega \right)\cdot \left(1-\frac{{\mathrm{Pr}}_{\bar{F}}\left(\omega \right)}{{\mathrm{Pr}}_{\bar{G}}\left(\omega \right)}\right)\\ \; & \leq \sum_{\omega \in \Omega_{>}\cap \Omega_{\mathrm{bad}}}{\mathrm{Pr}}_{\bar{G}}\left(\omega \right)+\epsilon \sum_{\omega \in \Omega_{>}\setminus \Omega_{\mathrm{bad}}}{\mathrm{Pr}}_{\bar{G}}\left(\omega \right)\\ \; & \leq \mathrm{Pr}[\left(\tau \left(A^{G}\right),S^{\prime }\right)\in \Omega_{\mathrm{bad}}]+\epsilon . \end{array}$$
□

### 4.4. Expectation Method

Hoang and Tessaro [82] introduced a somewhat generalized version of the H-technique, termed the *expectation method*. We describe it in a slightly different way in order to conform to our notation.

**Lemma** **4**(Expectation Method)**.** *Suppose that*
$\bar{F}:=\left(F,S\right)$
*and*
G¯:=(G,S′)
*are two*
S*-extended*
(X,Y)
*response systems. Let *Ω* be the support of*
${\mathrm{Pr}}_{\bar{G}}$
*and suppose that there is a set*
$\Omega_{\mathrm{bad}}\subseteq \Omega$*, and a non-negative function*
$\epsilon :X^{q}\times Y^{q}\times S\to \left[0,\infty \right)$
*such that*
$\forall \bar{\tau }=\left(x^{q},y^{q},s\right)\notin \Omega_{\mathrm{bad}}$*, we have*
$$\frac{\mathrm{Pr}[F\left(x^{q}\right)=y^{q},S=s]}{\mathrm{Pr}[G\left(x^{q}\right)=y^{q},S^{\prime }=s]}\geq 1-\epsilon \left(\bar{\tau }\right).$$
*Then, for any*
(X,Y)
*adversary*
A*,*
(8)$$\begin{array}{cc} \Delta_{A}\left(F\;;\;G\right) & \leq \mathrm{Pr}\left[\left(\tau \left(A^{G}\right),S^{\prime }\right)\in \Omega_{\mathrm{bad}}\right]+\mathrm{Ex}\left[\epsilon \left(\bar{\tau }\left(A^{G}\right)\right)\right]. \end{array}$$

One can set $\epsilon (\bar{\tau })=1$, for $\bar{\tau }\in \Omega_{\mathrm{bad}}$ to avoid the separate calculation of the bad transcript probability. The extended H-technique is obtained using Equation (Equation 8), when ϵ is a constant function. Based on CS’s view of the H-technique, the expectation method is like partitioning the set of transcripts into singletons. Thus, one could argue that the expectation method should achieve optimality. This is possible if one could identify a suitable definition of the ϵ function and provide a tight estimation of the expectation value. Specifically, for $\bar{\tau }\in X^{q}\times Y^{q}\times S$, we define $\epsilon (\bar{\tau })$ as
$$\epsilon \left(\bar{\tau }\right)=\left\{\begin{array}{cc} 1-\frac{{\mathrm{Pr}}_{\bar{F}}\left(x^{q},y^{q},s\right)}{{\mathrm{Pr}}_{\bar{G}}\left(x^{q},y^{q},s\right)} & \mathrm{when}\;{\mathrm{Pr}}_{\bar{G}}\left(x^{q},y^{q},s\right)>{\mathrm{Pr}}_{\bar{F}}\left(x^{q},y^{q},s\right),\\ 0 & \mathrm{otherwise}. \end{array}\right.$$
Now equality holds in Equation (Equation 8), if we apply the expectation method with $\Omega_{\mathrm{bad}}=\emptyset$.

## 5. Hash-Based Constructions

Now we briefly describe the power of the (extended) H-technique by proving the security of some hash-based constructions. A hash function $H:M\overset{*}{\to }X$ is called *ϵ-universal* if for all $m\neq m^{\prime }\in M$, ${\mathrm{Pr}}_{H\leftarrow_{\$}H}\left[H\left(m\right)=H\left(m^{\prime }\right)\right]=\leq \epsilon$.

### 5.1. Hash-Then-PRF

Construction. Let $H:M\overset{*}{\to }X$ be an ϵ-universal hash and $\rho :X\overset{*}{\to }Y$ be a random function. The composition function $\rho \circ H:M\overset{*}{\to }Y$ is known as the hash-then-PRF construction (see Figure 1). This construction has been studied in [102,103]. Many PRF constructions can be viewed as hash-then-PRF constructions. For example, EMAC [104], ECBC and FCBC [15], LightMAC [105], and the protected counter sum or PCS approach [43].

**Lemma** **5.** 
*Let hash-then-PRF be defined as above. Then, we have*

Advρ∘Hprf(q)≤q2ϵ.



**Proof.** We recall that all adversaries considered in this paper are deterministic and make no redundant queries (in this case, all queries are distinct). The basic idea of the proof is that as long as there is no collision among the hash outputs, ρ returns random values and hence the composition function behaves like a random function defined over a larger input space M. We capture this to prove the lemma formally using the extended H-technique.Extended Systems. We denote the composition system F=ρ∘H. Let $\rho^{\prime }$ be a random function from the message space M to Y. We denote the size of the set Y as *N*. Let H be the key space of the hash function. We define a H-extended random system. In the ideal system $\rho^{\prime }$, we simply adjoin a hash key $H\leftarrow_{\$}H$ chosen independently of ρ. Let ${\bar{\rho }}^{\prime }=\left(\rho ,H\right)$ be the extended system. In the case of F, we simply release the hash key $H^{\prime }$. We denote the extended system $\bar{F}=\left(F,H\right)$.Bad Transcripts and Its Analysis. Let $\tau =(m^{q},y^{q},h)$ denote a transcript, where $m^{q}\in M^{\left(q\right)}$, $y^{q}\in Y^{q}$, and h∈H. We define $x^{q}=h\left(m^{q}\right):=H\left(h,m^{q}\right)$ (i.e., *h* is the key of H). As mentioned above, we say that a transcript is bad if
there exist distinct i,j∈[q], such that $x_{i}=x_{j}$.We bound the probability of this event by q2ϵ, as the hash function is ϵ-universal and there are, at most, q2(i,j) pairs. All other transcripts are considered to be good.Good Transcript Analysis. Fix a good transcript $\tau =(m^{q},y^{q},h)$. In the ideal world, we have
$$\mathrm{Pr}\left[\rho^{\prime }\left(m^{q}\right)=y^{q},H^{\prime }=h\right]=\frac{1}{\left|H\right|}\times \frac{1}{N^{q}}.$$
In the real world, we have
$$\begin{array}{cc} \mathrm{Pr}[F\left(m^{q}\right)=y^{q},H=h] & =\frac{1}{\left|H\right|}\times \mathrm{Pr}\left[F\left(m^{q}\right)=y^{q}\;|\;H=h\right]\\ \; & =\frac{1}{\left|H\right|}\times \mathrm{Pr}\left[\rho \left(x^{q}\right)=y^{q}\right]\\ \; & =\frac{1}{\left|H\right|}\times \frac{1}{N^{q}}\\ \; & =\mathrm{Pr}[\rho^{\prime }\left(m^{q}\right)=y^{q},H=h]. \end{array}$$
The result follows, using the extended H-technique.□

### 5.2. Hash-Then-TBC

Construction. Let $H:=\left(H_{1},H_{2}\right):M\overset{*}{\to }T\times X$ be an ϵ-universal hash such that $H_{1}$ is an $\epsilon_{1}$-universal hash. Let $\tilde{\pi }$ be a tweakable random permutation on X with a tweak space T. We define the composition function $F=\tilde{\pi }\circ H$ as hash-then-TBC (see Figure 2).

A special instantiation (in which $H_{1}$ and $H_{2}$ are assumed to be independent) of the above construction is first considered in [63]. Subsequently, the analysis of the above construction was been performed [66]. In the same paper, the composition was used to define an MAC, called ZMAC+. Note that a tweakable random permutation is a PRF with a maximum advantage about $q^{2}/2N$, where *N* is the size of the set X (this is similar to the well known result of PRP-PRF switching lemma [3]). Thus, one can apply the previous result for this construction. However, the construction can be shown to have a better PRF advantage. Let us denote the size of T and X by *T* and *N*, respectively. Let ρ be an ideal candidate, i.e., a random function from M to X.

In the previous construction we avoided the collision among hash outputs since the hash outputs were fed into a random function. In this case, the hash output is fed into a tweakable random permutation (as an input as well as a tweak). Hence, we need to avoid simultaneous collisions ofthe tweak and output, as well as the tweak and input of $\tilde{\pi }$. The following lemma was proved in [66] using the H-technique. We first tackle this problem without extending the random system which would require slightly more effort and then show how the extended H-technique, as well as the expectation method, can help to bound the advantage very easily.

**Lemma** **6.** 
*Let Hash-then-TBC be defined as above. Then we have*

Advπ˜∘Hprf(q)≤q2ϵ+q2ϵ1N.



#### 5.2.1. A. Proof without Releasing the Internal Values

Let $\tau =(m^{q},y^{q})$ be the transcript at hand. Let $C:=C(y^{q})$ be the number of colliding pairs in the output tuple $y^{q}$. More formally,
$$C=|\{\left(i,j\right)\;:\;i<j,y_{i}=y_{j}\}|.$$
When $Y_{1},\ldots ,Y_{q}\leftarrow_{\$}X^{q}$, we write the random variable $C(Y^{q})$ as C.

Good Hash Key. Let H be the key space of the hash function. We define a subset $H_{\mathrm{good}}\subseteq H$ as the set of all h∈H so that there is
no collision among $\left(t_{1},x_{1}\right),\ldots ,\left(t_{q},x_{q}\right)$ andno collision among $\left(t_{1},y_{1}\right),\ldots ,\left(t_{q},y_{q}\right)$.
where $h\left(m^{q}\right)=\left(t^{q},x^{q}\right)$.

Clearly, for a good hash key *h*, $\left(t^{q},x^{q},y^{q}\right)$ is tweakable-permutation-compatible and $\mathrm{Pr}\left[\tilde{\pi }\left(t^{q},x^{q}\right)=y^{q}\right]\geq N^{-q}$. In the following, for any *h*, we denote $h\left(m^{q}\right)=\left(t^{q},x^{q}\right)$.
(9)Pr[F(mq)=yq]≥∑h∈HgoodPr[H=h,π˜(H(mq))=yq]=∑h∈HgoodPr[H=h,π˜(tq,xq)=yq]=∑h∈HgoodPr[H=h]×Pr[π˜(tq,xq)=yq]≥∑h∈HgoodPr[H=h]×N−q=1−Pr[H∉Hgood]×N−q≥1−q2·ϵ−C·ϵ1×N−q
The last inequality follows from the union bound. A bad hash key can arise due to either collision of tweak-input pairs (which happens with a probability of, at most, q2·ϵ) or collision of tweak-output pairs. As there are *C* pairs at which $y_{i}$ values collide, we must have collisions of tweak values among these *C* pairs. Hence, the probability of a tweak–output collision occurring is at most $C\cdot \epsilon_{1}$. This justifies the last inequality. Now, the ratio
Pr[F(Mq)=yq]Pr[ρ′(Mq)=yq]≥1−q2·ϵ−C·ϵ1.

To obtain a bound, it is necessary to obtain a good upper bound for $C(y^{q})$ for all $y^{q}$. At this point, we have two options to bound the value of $C(y^{q})$.

##### (1) Standard H-Technique:

In this case we can use Markov’s inequality to bound $C=C(Y^{q})$ to a moderate value, where $Y^{q}$ is a *q*-tuple independent uniform random variable (responses of ρ). We can write $C=\sum_{i<j}I_{i,j}$ where $I_{i,j}$ is the binary random variable which takes value the value of 1 if $Y_{i}\neq Y_{j}$. So,
$$\mathrm{Ex}\left(C\right)=\frac{q(q-1)}{2N}.$$
Bad Transcripts and Their Analysis. Let α be a threshold parameter (which is determined below). We call a transcript $\left(m^{q},y^{q}\right)$
*bad* if the number of collision pairs of $y_{i}$ values is greater than α. Using Markov’s inequality, we obtain
(10)Pr[τ(Aρ)∈Ωbad]=Pr[C≥α]≤E[C]/α=q2·1αN
for any adversary A.

Analysis of Good Transcripts. Now, fix any good transcript $\tau =(m^{q},y^{q})$. Using α as an upper bound for $C(y^{q})$ in Equation (Equation 9), we get
Pr[F(Mq)=yq]Pr[ρ′(Mq)=yq]≥1−q2·ϵ−α·ϵ1.

Finally, using the bad transcript probability of Equation (Equation 10) and the standard H-technique, we obtain
(11)Advπ˜∘Hprf≤q2·1αN+q2·ϵ+α·ϵ1.
By equating the two terms q2·1αN and $\alpha \cdot \epsilon_{1}$, we set $\alpha =\frac{q}{\sqrt{N\epsilon_{1}}}$. With this choice of α, the PRF advantage is bounded as
Advπ˜∘Hprf≤q2·ϵ+2qϵ1/N.□

##### (2) Expectation Method:

For small messages, there are universal hash functionswith ϵ≈1/NT and $\epsilon_{1}\approx 1/T$. In this case, the standard H-technique bounds the prf advantage to $O\left(q^{2}/NT\right)+O\left(q/\sqrt{NT}\right)$. Clearly, the dominating term is $O(q/\sqrt{NT})$. Instead of a crude estimation of C, if we apply the expectation method (which needs to work with the expected value of C instead of an upper bound), we can get rid of the dominating term $O(q/\sqrt{NT})$.

We define $\epsilon :M^{q}\times X^{q}\to \left[0,\infty \right)$ via the mapping
ϵ(τ)=q2ϵ+C·ϵ1.
Clearly ϵ is non-negative and the ratio of real to ideal interpolation probabilities is at least 1−ϵ(τ) (using Equation (Equation 9)). Thus, we can use Lemma 4 to get
(12)Advπ˜∘Hprf(q)≤Ex[ϵ(τ)]=q2ϵ+q2ϵ1N.

#### 5.2.2. B. Proof by Releasing the Internal Values

Now we show that the extended H-technique can also help to provide a bound for this construction very easily.

Extended Systems. Let H be the key space of the hash function. We define the H-extended random system. In the ideal system ρ, we simply adjoin a hash key $H\leftarrow_{\$}H$ chosen independently of ρ. Let $\bar{\rho }=\left(\rho ,H\right)$ be the extended system. In case of a F value based on the hash key $H^{\prime }$ and tweakable random permutation $\tilde{\pi }$, we release the hash key $H^{\prime }$. We denote the extended system $\bar{F}=\left(F,H^{\prime }\right)$.

Bad Transcripts and Their Analysis. Given any hash key *h*, we define $h\left(m^{q}\right)=\left(t^{q},x^{q}\right)$. We can state that an extended transcript $\left(m^{q},y^{q},h\right)$ is bad if either
there is a collision among $\left(t^{q},x^{q}\right)$ orthere is a collision among $\left(t^{q},y^{q}\right)$.

In the extended ideal world, an adversary can realize a bad transcript with a probability of, at most, q2·(ϵ+ϵ1/N). The probability that there is a collision among $\left(t^{q},x^{q}\right)$ is, at most, q2·ϵ. The probability that there is a collision among $\left(t^{q},y^{q}\right)$ is, at most, q2ϵ1N (a pair of *y* values will collide with a probability of 1/N, whereas a pair of *t* values will collide with a probability of, at most, $\epsilon_{1}$ and these events are independent).

Analysis of a Good Transcript. Now we fix a good transcript $\tau =(m^{q},y^{q},h)$. For the ideal system we have,
$$\mathrm{Pr}\left[\bar{\rho }\left(m^{q}\right)=\left(y^{q},h\right)\right]=\frac{1}{\left|H\right|}\times \frac{1}{N^{q}},$$
and for the real system we have,
$$\mathrm{Pr}\left[\bar{F}\left(m^{q}\right)=\left(y^{q},h\right)\right]=\mathrm{Pr}\left[H=h\right]\times \mathrm{Pr}\left[\tilde{\pi }\left(t^{q},x^{q}\right)=y^{q}\right]\geq \frac{1}{\left|H\right|}\times \frac{1}{N^{q}}.$$
Note that we have applied Equation (Equation 4) for the real system, as the transcript is tweakable-permutation-consistent and it is non-redundant. Hence, the extended H-technique of Lemma 3 gives,
(13)Advπ˜∘Hprf(q)≤q2·(ϵ+ϵ1/N).

**Remark** **1.** 
*The*

*XTX*

*[81] and HaT [63] constructions are quite similar to HtTBC [66]. Consequently, we obtain similar proofs for these constructions.*


### 5.3. An Extension of Naor–Reingold

The basic version of ENR [78] is a 2n-bit permutation based on an (n,n)-TBC and an *n*-bit AXU hash function, essentially adapting the Naor–Reingold [106] simplification of the four-round Feistel structure. Here, we describe a version which generalizes ENR (see Figure 3) based on (t,n)-TBC (for t≤n), as well as LDT [58], in which the hash function is not present (which can also be viewed as an identity function).

Construction. Let $M=F_{2^{t}}\times F_{2^{n}}$. Suppose that $H:=\left(H_{1},H_{2}\right):M\overset{*}{\to }F_{2^{t}}\times F_{2^{n}}$ is an invertible keyed function such that $H_{1}$ is an ϵ-universal hash function. Suppose that ${\tilde{\pi }}_{1}$ and ${\tilde{\pi }}_{2}$ are two independently sampled tweakable random permutations on $F_{2^{n}}$ with tweak space $F_{2^{t}}$. We define ${\mathrm{NR}}^{⋆}:F_{2^{n}}\times F_{2^{t}}\overset{*}{\to }F_{2^{n}}\times F_{2^{t}}$ as follows:
–Input: m∈M.
(v,u)=H(h,m).$x∥y={\tilde{\pi }}_{1}\left(v,u\right)$, where $x\in F_{2^{t}}$ and $y\in F_{2^{n-t}}$.$z={\tilde{\pi }}_{2}\left(x,y∥v\right)$.$c=H^{-1}\left(h,x∥z\right)$.return $c:={\mathrm{NR}}^{⋆}(m)$.

Let F be the response system corresponding to ${\mathrm{NR}}^{⋆}$ and Π be the system corresponding to a random permutation over $F_{2^{n}}\times F_{2^{t}}$. It is easy to see that F is an invertible response system.

**Lemma** **7.** 

AdvNR⋆sprp(q)≤q22nϵ+12t.



**Proof.** We apply extended H-technique and so we define the additional random variables released after the interaction.Extended Systems. Suppose that $\left(\delta^{q},m^{q},c^{q}\right)$ is the forward-only transcript (before we extend it). Now, we define the $\left(H\times F_{2^{n-t}}^{q}\right)$-extended random system. In the ideal system Π, we simply adjoin a hash key $H\leftarrow_{\$}H$ and $Y_{1},\ldots ,Y_{q}\leftarrow_{\$}F_{2^{n-t}}$, chosen independently of Π. Let $\bar{\Pi^{\pm }}=\left(\Pi^{\pm },H,Y^{q}\right)$ be the extended ideal system.In the case of $F^{\pm }$, based on the hash key H and tweakable random permutation ${\tilde{\pi }}_{1},{\tilde{\pi }}_{2}$, we release the hash key H and all *q* internal values $Y_{1},\ldots ,Y_{q}$, where $Y_{i}$ is the value of *y* in step-2, while computing $F(m_{i})$. We denote the extended system $\bar{F^{\pm }}=\left(F^{\pm },H,Y^{q}\right)$.Analysis of Bad Transcripts. Let $\tau =(\delta^{q},m^{q},c^{q},h,y^{q})$ be a transcript. We define $h\left(m^{q}\right)=\left(v^{q},u^{q}\right)$ and $h\left(c^{q}\right)=\left(x^{q},z^{q}\right)$. We can say that an extended transcript τ is bad if there is a collision among $v_{1}∥x_{1}∥y_{1},v_{2}∥x_{2}∥y_{2},\ldots ,v_{q}∥y_{q}∥x_{q}$ values. (Observe that the bad event is quite similar to the one arising in hash-then-TBC analysis. In fact, in most of the TBC-based constructions, the sole bad event is of this particular type (avoiding tweak-input and tweak-output collisions).)Now we calculate the probability that the extended transcript $\bar{\tau }\left(A^{\Pi^{\pm }}\right)$ is bad for any adversary A making *q* queries. Let ${\left[q\right]}_{e}$ and ${\left[q\right]}_{d}$ denote the set of all forward- and backward-query indices. Let us denote the random variables corresponding to m,c,x,y, and *v* values in the ideal world as M,C,X,Y, and V, respectively.Now, the bad event means that there is i<j such that $X_{i}=X_{j},Y_{i}=Y_{j},V_{i}=V_{j}$. We have $\mathrm{Pr}\left[Y_{i}=Y_{j}\right]=2^{t-n}$. Moreover, $Y_{i}$ values are chosen independently of (Π,H) and are hence independent of the values of X and V. Thus, it is sufficient to bound $\mathrm{Pr}[X_{i}=X_{j},V_{i}=V_{j}]$ for some i<j.**Claim**.
$$\mathrm{Pr}\left[X_{i}=X_{j},V_{i}=V_{j}\right]\leq \frac{\epsilon }{2^{t}}.$$
We prove the claim when $j\in {\left[q\right]}_{e}$. A similar proof is applied for $j\in {\left[q\right]}_{d}$. As $j\in {\left[q\right]}_{e}$, $V_{j}$ depends on $C_{j}$ and the hash key H of H. We first condition on all query responses $M^{j-1}=m^{j-1},C^{j-1}=c^{j-1}$ up to j−1 queries. Note that up to j−1 queries, the queries can be both encryption or decryption queries. Thus, $M^{j-1},C^{j-1}$ is simply the forward-only reordering of the query and responses. Once we condition on itand $j\in {\left[q\right]}_{e}$, the value of $M_{j}$ is fixed (say, $m_{j}$) and $C_{j}\leftarrow_{\$}M\setminus \left\{c_{1},\ldots ,c_{j-1}\right\}$. Let us write the conditional event $M^{j-1}=m^{j-1},C^{j-1}=c^{j-1}$ as E and the set of all *h* for which $H_{1}\left(h,m_{i}\right)=H_{1}\left(h,m_{j}\right)$ holds as $H^{\prime }$. Thus,
$$\begin{array}{cc} \mathrm{Pr}[X_{i}=X_{j},V_{i}=V_{j}\;|\;E] & =\mathrm{Pr}[H_{1}\left(H,m_{i}\right)=H_{1}\left(H,m_{j}\right),\;H_{1}\left(H,c_{i}\right)=H_{1}\left(H,C_{j}\right)\;|\;E]\\ \; & =\sum_{h\in H^{\prime }}\mathrm{Pr}\left[H=h,\;H_{1}\left(h,c_{i}\right)=H_{1}\left(h,C_{j}\right)\;|\;E\right]\\ \; & =\sum_{h\in H^{\prime }}\mathrm{Pr}\left[H=h\right]\times \mathrm{Pr}\left[H_{1}\left(h,c_{i}\right)=H_{1}\left(h,C_{j}\right)\;|\;E\right]\\ \; & \leq \sum_{h\in H^{\prime }}\mathrm{Pr}\left[H=h\right]\times \frac{1}{2^{t}} \end{array}$$
To justify the latter inequality, we first note that H(h,·) is an invertible function and so the conditional distribution of $H(h,C_{j})$ is uniformly distributed over a set of size $2^{n+t}-\left(j-1\right)$. Hence, $\mathrm{Pr}\left[H_{1}\left(h,c_{i}\right)=H_{1}\left(h,C_{j}\right)\;|\;E\right]\leq \frac{2^{n}}{2^{n+t}-\left(j-1\right)}\leq \frac{1}{2^{t}}$. To complete the proof of the claim we sum over all such events E (i.e., varying $m^{j-1}$ and $c^{j-1}$) after multiplying the probability of E.Therefore, for any i<j, $\mathrm{Pr}\left[X_{i}=X_{j},V_{i}=V_{j},Y_{i}=Y_{j}\right]\leq \frac{\epsilon }{2^{n}}$. This proves that
(14)$$\mathrm{Pr}\left[\bar{\tau }\left(A^{\Pi^{\pm }}\right)isbad\right]\leq \frac{q(q-1)\epsilon }{2^{n+1}}.$$
Analysis of a Good Transcript. We fix a good transcript $\tau =(\delta^{q},m^{q},c^{q},h,y^{q})$ and let $h\left(m^{q}\right)=\left(v^{q},u^{q}\right)$ and $h\left(c^{q}\right)=\left(x^{q},z^{q}\right)$. According to the definition of a good transcript, $\left(v^{q},u^{q},x^{q}∥y^{q}\right)$ and $\left(x^{q},y^{q}∥v^{q},z^{q}\right)$ are tweakable-permutation-compatible, which means they are also non-redundant. Thus,
$$\begin{array}{cc} \mathrm{Pr}[F\left(m^{q}\right)=c^{q},H=h,Y^{\prime q}=y^{q}] & =\mathrm{Pr}\left[{\tilde{\pi }}_{1}\left(v^{q},u^{q}\right)=x^{q}∥y^{q}\right]\times \mathrm{Pr}\left[{\tilde{\pi }}_{2}\left(x^{q},y^{q}∥v^{q}\right)=z^{q}\right]\times \mathrm{Pr}\left[H=h\right]\\ \; & \geq \frac{1}{\left|H\right|}\times \frac{1}{2^{2nq}}. \end{array}$$
On the other hand, realizing the transcript via the extended ideal system is expressed as
$$\begin{array}{cc} \mathrm{Pr}[\Pi \left(m^{q}\right)=c^{q},H=h,Y^{q}=y^{q}] & =\frac{1}{\left|H\right|}\times \frac{1}{{\left(2^{n+t}\right)}_{q}}\times \frac{1}{2^{(n-t)q}}\\ \; & \leq \frac{1}{\left|H\right|}\times \frac{1}{2^{(n+t)q}\cdot \left(1-\frac{q(q-1)}{2^{n+t+1}}\right)}\times \frac{1}{2^{(n-t)q}}. \end{array}$$
Thus, the ratio
(15)Pr[F(mq)=cq,H=h,Y′q=yq]Pr[Π(mq)=cq,H=h,Yq=yq]≥1−q22n+t.
Combining Equations (Equation 14) and (Equation 15) with Lemma 3, we get
AdvNR⋆sprp(q)≤q22nϵ+12t.□

Based on the security analysis of this generic design, we can obtain simple proofs for ENR [78,79] and LDT [58].

#### 5.3.1. A Simple Proof for ENR

The basic version of ENR [78] can be viewed as a specific instantiation of NR*, where the hash function is defined as (a,b)↦(a,(a⊙K⊕b)) for $K\leftarrow_{\$}F_{2^{n}}$. Subsequently, Minematsu and Iwata presented a simpler definition for ENR for t<n, called SmallBlock [79] that merely redefines the hash to be ${\left(a,\left(a⊙K⊕b\right)\right)}_{|t}$. Now, we have the following corollary.

**Corollary** **2.** 
*For*

t≤n

*we have*

$${\mathrm{Adv}}_{\mathrm{ENR}}^{\mathrm{sprp}}\left(q\right)\leq \frac{q^{2}}{2^{n+t}}.$$



#### 5.3.2. A Simple Proof for LDT

The two-round LDT construction by Chen et al. [58] can also be viewed as a specific instantiation of NR*, where the hash function is defined to be the identity function. This immediately gives the following corollary on the SPRP advantage of LDT.

**Corollary** **3.** 

$${\mathrm{Adv}}_{\mathrm{LDT}}^{\mathrm{sprp}}\left(q\right)\leq \frac{q^{2}}{2^{n}}.$$



## 6. Feistel Structure-Based Schemes

A (keyed) bijective function Ψ based on an internal primitive ψ is said to be *inverse-free* if and only if the computation of $\Psi^{-1}$ does not require the execution of $\psi^{-1}$. The Feistel structure has this property.

### 6.1. Three-Round Luby–Rackoff

Construction. Suppose that ψ is a random function over ${\left\{0,1\right\}}^{n}$ and Ψ is a round function defined by the mapping
$$\left(a,b\right)\overset{\Psi }{\mapsto }\left(b,a⊕\psi \left(b\right)\right).$$
Suppose that $\Psi_{i}$ denotes the round function based on the random function $\psi_{i}$. The well-known three-round Luby–Rackoff [2] scheme (see Figure 4), denoted LR3, is defined as $\Psi_{3}\circ \Psi_{2}\circ \Psi_{1}$.

LR3 is a well-studied birthday-bound pseudorandom permutation. The original proof by Luby and Rackoff [2] is one of the foundational results in symmetric-key provable security. We now show how a fairly modern tool in symmetric-key provable security can simplify the security analysis as compared to the original proof. We note that a simpler proof based on the H-technique is already available through the work of Nachef, Patarin, and Volte [107]. Here, we provide the proof of LR3 in our language.

**Lemma** **8.** 
*For*

t≤n

*, we have*

$${\mathrm{Adv}}_{3\mathrm{LR}}^{\mathrm{prp}}\left(q\right)\leq \frac{q^{2}}{2^{n}}+\frac{q^{2}}{2^{2n}}.$$



**Proof.** We apply the standard H-technique (i.e., no need to extend the system).Analysis of Bad Transcripts. For input (a,b) and output (c,d), let the one-round and two-round outputs be (x,b) and (x,d). Let F be the response system corresponding to LR3 and Π be the system corresponding to a random permutation. The transcript random variable τ is defined as the tuple $\left(A^{q},B^{q},C^{q},D^{q}\right)$. We say that a transcript $\left(a^{q},b^{q},c^{q},d^{q}\right)$ is bad if $d^{q}$ has a colliding pair, i.e., for two distinct queries *i* and *j*, $d_{i}=d_{j}$. Thus we have
(16)$$\mathrm{Pr}\left[\tau \left(A^{\Pi }\right)\;\mathrm{is}\;\mathrm{bad}\right]\leq \frac{q\left(q-1\right)\left(2^{n}-1\right)}{2(2^{2n}-1)}\leq \frac{q^{2}}{2^{n+1}}.$$
Analysis of Good Transcripts. Fix a good transcript $\left(a^{q},b^{q},c^{q},d^{q}\right)$. We say that a function f∈Func is bad, denoted $f\in {\mathrm{Func}}_{\mathrm{bad}}$ if for some distinct i,j∈[q],$f\left(b_{i}\right)⊕a_{i}=f\left(b_{j}\right)⊕a_{j}$; otherwise, we say it is good. Clearly for a uniform random *f*, the probability of *f* being bad is bounded to, at most, $q^{2}/2^{n+1}$. Let ${\mathrm{Func}}_{\mathrm{good}}=\mathrm{Func}\setminus {\mathrm{Func}}_{\mathrm{bad}}$. Thus we have
$$\begin{array}{cc} \mathrm{Pr}[F\left(a^{q},b^{q}\right)=\left(c^{q},d^{q}\right)] & \geq \mathrm{Pr}\left[\psi_{1}\in {\mathrm{Func}}_{\mathrm{good}}\right]\times \mathrm{Pr}\left[\psi_{2}\left(x^{q}\right)=b^{q}⊕d^{q},\psi_{3}\left(d^{q}\right)=x^{q}⊕c^{q}\;|\;\psi_{1}\right]\\ \; & \geq \left(1-\frac{q^{2}}{2^{n+1}}\right)\times \frac{1}{2^{2nq}}\\ \; & \geq \left(1-\frac{q^{2}}{2^{n+1}}-\frac{q^{2}}{2^{2n}}\right)\times \frac{1}{{\left(2^{2n}\right)}_{q}} \end{array}$$
As $\mathrm{Pr}\left[\Pi \left(a^{q},b^{q}\right)=\left(c^{q},d^{q}\right)\right]=\frac{1}{{\left(2^{2n}\right)}_{q}}$, we have
(17)$$\frac{\mathrm{Pr}[F\left(a^{q},b^{q}\right)=\left(c^{q},d^{q}\right)]}{\mathrm{Pr}[\Pi \left(a^{q},b^{q}\right)=\left(c^{q},d^{q}\right)]}\geq \left(1-\frac{q^{2}}{2^{n}}-\frac{q^{2}}{2^{2n}}\right).$$The result follows from Equations (Equation 16) and (Equation 17) using the standard H-technique. □

**Remark** **2.** 
*Note that the above proof can be easily converted into a proof with an extended transcript. In particular, we release*

$X^{q}$

*values. In the case of an ideal oracle, it is computed as follows:*

$X_{i}=\psi_{1}\left(b_{i}\right)⊕a_{i}$

*for all i. We add one more bad event, which is the presence of a collision among*

$X^{q}$

*values. The probability of this new bad event can be easily shown to be at most*

$q^{2}/2^{n+1}$

*. For a good transcript the ratio can be similarly shown to be at least*

$1-q^{2}/2^{2n}$

*and hence we obtain exactly the same bound for the PRP advantage.*

*One can have an SPRP proof for the four-round LR using the extended transcript. The proof is similar, with some bad events avoiding all possible collisions among inputs of*

$\psi_{2}$

*and*

$\psi_{3}$

*.*


### 6.2. Three-Round TBC-Based Luby–Rackoff

In [95], Coron et al. presented an alternative to the three-round Luby–Rackoff method using a tweakable block cipher, called TLR3, and demonstrated $O(q/2^{n})$-query security. In this case, ψ is a tweakable random permutation and the Ψ function is defined by the mapping $\left(a,b\right)\overset{\Psi }{\mapsto }\left(b,\psi \left(b,a\right)\right)$. The original work by Coron et al. is mainly focused on the indifferentiability of TLR3 with respect to an ideal cipher. However, their result also implies $\Omega (2^{n})$-query SPRP security. We present a relatively simple proof for the SPRP security of TLR3.

**Proposition** **1.** 
*For*

$q<2^{n-1}$

*we have*

$${\mathrm{Adv}}_{\mathrm{TLR}3}^{\mathrm{sprp}}\left(q\right)\leq \frac{q^{2}}{2^{2n-1}}.$$



**Proof.** We will use the extended H-technique to prove the claimed security.Extended Systems. The variables arising in the following analysis are analogous to the ones given in Figure 5. Let F be the response system corresponding to TLR3 and Π be the system corresponding to a random permutation. The transcript τ is defined as the tuple $\left(A^{q},B^{q},C^{q},D^{q}\right)$. We define $F_{2^{n}}^{q}$-extended response systems by adjoining the internal value $X^{q}$. In the case of F, this is well-defined according to the definition of TLR3.In the ideal system Π, we sample $X^{q}$ as follows:
for all $i\in {\left[q\right]}_{e}$,
$$X_{i}\leftarrow_{\$}{\left\{0,1\right\}}^{n}\setminus \left\{x\in {\left\{0,1\right\}}^{n}:\exists j<i,X_{j}=x\wedge B_{i}=B_{j}\right\};$$for all $i\in {\left[q\right]}_{d}$,
$$X_{i}\leftarrow_{\$}{\left\{0,1\right\}}^{n}\setminus \left\{x\in {\left\{0,1\right\}}^{n}:\exists j<i,X_{j}=x\wedge D_{i}=D_{j}\right\};$$Bad Transcripts and Their Analysis. We say that an extended transcript $\left(a^{q},b^{q},x^{q},c^{q},d^{q}\right)$ is bad if and only if $\left(b^{q},a^{q},x^{q}\right)$, $\left(x^{q},b^{q},d^{q}\right)$ and $\left(d^{q},x^{q},c^{q}\right)$ are not tweakable-permutation-consistent. Due to the way in which we sample $X^{q}$ in Π, the necessary and sufficient condition for the inconsistency of $\left(B^{q},A^{q},X^{q}\right)$, $\left(X^{q},B^{q},D^{q}\right)$, and $\left(D^{q},X^{q},C^{q}\right)$ is: i<j∈[q] and (1) $j\in {\left[q\right]}_{e}$ and $\left(X_{j},D_{j}\right)=\left(X_{i},D_{i}\right)$; or (2) $j\in {\left[q\right]}_{d}$ and $\left(B_{j},X_{j}\right)=\left(B_{i},X_{i}\right)$. Formally, we have
(18)Pr[τ¯(Π)∈Vbad]=∑i∈[q]∑i<j∈[q]ePr[Xi=Xj,Di=Dj]+∑i<j∈[q]dPr[Bi=Bj,Xi=Xj]≤q2×2n−122n−j+1×12n−q≤q(q−1)2n+1(2n−q).
Analysis of Good Transcripts. For a good transcript $\left(a^{q},b^{q},x^{q},c^{q},d^{q}\right)$, we know that $\left(b^{q},a^{q},x^{q}\right)$, $\left(x^{q},b^{q},d^{q}\right)$, and $\left(d^{q},x^{q},c^{q}\right)$ are tweakable-permutation-consistent. Let $\alpha^{u}=\mathrm{mcoll}\left(b^{q}\right)$, $\beta^{v}=\mathrm{mcoll}\left(x^{q}\right)$, and $\gamma^{w}=\mathrm{mcoll}\left(d^{q}\right)$. Given a good transcript, for a real system we have
$$\begin{array}{cc} \mathrm{Pr}[\bar{\tau }\left(F\right)] & =\mathrm{Pr}\left[\psi_{1}\left(b^{q},a^{q}\right)=x^{q}\right]\times \mathrm{Pr}\left[\psi_{2}\left(x^{q},b^{q}\right)=d^{q}\right]\times \mathrm{Pr}\left[\psi_{3}\left(d^{q},x^{q}\right)=c^{q}\right]\\ \; & =\frac{1}{\prod_{i=1}^{u}{\left(2^{n}\right)}_{\alpha_{i}}}\times \frac{1}{\prod_{j=1}^{v}{\left(2^{n}\right)}_{\beta_{j}}}\times \frac{1}{\prod_{k=1}^{w}{\left(2^{n}\right)}_{\gamma_{k}}}. \end{array}$$
For i∈[q], let $r_{i}$ and $s_{i}$ denote the number of previous queries *j* and $j^{\prime }$ such that $b_{i}=b_{j}$ and $d_{i}=d_{j^{\prime }}$, respectively. Then, for the ideal system we have
$$\begin{array}{cc} \mathrm{Pr}[\bar{\tau }\left(\Pi \right)] & =\frac{1}{{\left(2^{2n}\right)}_{q}}\times \frac{1}{\prod_{i^{\prime }\in {\left[q\right]}_{e}}\left(2^{n}-r_{i^{\prime }}\right)}\times \frac{1}{\prod_{k^{\prime }\in {\left[q\right]}_{d}}\left(2^{n}-s_{k^{\prime }}\right)}\\ \; & \leq \frac{1}{\left(1-\frac{q^{2}}{2^{2n}}\right)\times 2^{2nq}}\times \frac{1}{\prod_{i^{\prime }\in {\left[q\right]}_{e}}\left(2^{n}-r_{i^{\prime }}\right)}\times \frac{1}{\prod_{k^{\prime }\in {\left[q\right]}_{d}}\left(2^{n}-s_{k^{\prime }}\right)}. \end{array}$$
Thus the ratio is
(19)$$\begin{array}{cc} \frac{\mathrm{Pr}[\bar{\tau }\left(F\right)]}{\mathrm{Pr}[\bar{\tau }\left(\Pi \right)]} & \geq \left(1-\frac{q^{2}}{2^{2n}}\right)\times \frac{2^{2nq}\times \prod_{i^{\prime }\in {\left[q\right]}_{e}}\left(2^{n}-r_{i^{\prime }}\right)\times \prod_{k^{\prime }\in {\left[q\right]}_{d}}\left(2^{n}-s_{k^{\prime }}\right)}{\prod_{i=1}^{u}{\left(2^{n}\right)}_{\alpha_{i}}\times \prod_{j=1}^{v}{\left(2^{n}\right)}_{\beta_{j}}\times \prod_{k=1}^{w}{\left(2^{n}\right)}_{\gamma_{k}}}. \end{array}$$
In the above expression, we claim the following:
$$\begin{array}{cc} \prod_{i=1}^{u}{\left(2^{n}\right)}_{\alpha_{i}} & =\prod_{i^{\prime }\in {\left[q\right]}_{e}}\left(2^{n}-r_{i^{\prime }}\right)\times \prod_{\hat{i}\in {\left[q\right]}_{d}}\left(2^{n}-{\hat{\gamma }}_{\hat{i}}\right),\;\mathrm{and}\\ \prod_{k=1}^{w}{\left(2^{n}\right)}_{\gamma_{k}} & =\prod_{k^{\prime }\in {\left[q\right]}_{d}}\left(2^{n}-s_{k^{\prime }}\right)\times \prod_{\hat{k}\in {\left[q\right]}_{e}}\left(2^{n}-{\hat{\gamma }}_{\hat{k}}\right), \end{array}$$
where for all $\hat{i}$ and $\hat{k}$, ${\hat{\gamma }}_{\hat{i}},{\hat{\gamma }}_{\hat{k}}\geq 0$. We argue the first one and the second can be argued similarly. The set [u] can be viewed as an indexing over the set of distinct tweak values. Now consider the first term on the right hand side (the one indexed by $i^{\prime }$). For all $i^{\prime }\in {\left[q\right]}_{e}$, we define $\phi \left(i^{\prime }\right)\to \left(i,p\right)$ such that *i* is the index of the tweak of the $i^{\prime }$-th query, i.e., i≤u, and *p* is the number of previous queries with the same tweak, i.e., $p=r_{i^{\prime }}\leq \alpha_{i}$. The mapping is well-defined. Furthermore, it is injective: for distinct $i_{1}^{\prime },i_{2}^{\prime }\in {\left[q\right]}_{e}$, either the tweaks are different, i.e., $i_{1}\neq i_{2}$, or if the tweaks are same, then $p_{1}=r_{i_{1}^{\prime }}\neq r_{i_{2}^{\prime }}=p_{2}$. Observe that ϕ also maps each of the $\left(N-r_{i^{\prime }}\right)$ terms on the right-hand side to a unique (N−p) term (taken from ${\left(N\right)}_{\alpha_{i}}$ expansion) on the left, exhausting all the terms corresponding to encryption queries. Thus, we are left with only the terms corresponding to all the decryption queries. Using the relations presented above in Equation (Equation 19) we have
(20)$$\begin{array}{cc} \frac{\mathrm{Pr}[\bar{\tau }\left(F\right)]}{\mathrm{Pr}[\bar{\tau }\left(\Pi \right)]} & \geq \left(1-\frac{q^{2}}{N^{2}}\right)\times \frac{N^{2q}}{\prod_{\hat{i}\in {\left[q\right]}_{d}}\left(N-{\hat{\gamma }}_{\hat{i}}\right)\times \prod_{j=1}^{v}{\left(N\right)}_{\beta_{j}}\times \prod_{\hat{k}\in {\left[q\right]}_{e}}\left(N-{\hat{\gamma }}_{\hat{k}}\right)}\\ \; & \geq \left(1-\frac{q^{2}}{N^{2}}\right) \end{array}$$
The result follows from the extended H-technique, using Equations (Equation 18) and (Equation 20). □

## 7. Strong Pseudo-Random Permutation Designs

More notations: For ℓ≥1, $p\in M^{\ell }$, p[i] denotes the *i*-th coordinate of *p*, for all i∈[ℓ]. Extending the notation, p[i..j] denotes the M-substring (p[i],p[i+1],…,p[j]) of *p*, for 1≤i<j≤ℓ. Thus, we also write *p*, alternatively, as p[1..ℓ]. We denote the set of all *n*-bits as $B_{n}$.

### 7.1. HCTR

HCTR is an encryption scheme developed by Wang, Feng and Fu [22], based on the hash-CTR-hash paradigm, which uses a sandwich consisting of the CTR mode in between two executions of an AXU hash function. The CTR mode can be replaced by a pseudorandom function (PRF), which takes *n*-bit inputs and returns an arbitrarily long bit-stream. The PRF-based HCTR construction was studied by Chakraborty et al. in [108], and this system does not require any inverse of the block cipher. For the sake of simplicity, we first describe a simple version of the HCTR construction.

Construction. Let *L* (≥n) be a large integer, which is the size of the largest messages to be encrypted. Let T denote a tweak space. Suppose that $H:T\times {\left\{0,1\right\}}^{\leq L-n}\to {\left\{0,1\right\}}^{n}$ is an ϵ-AXU hash function, π is an *n*-bit random permutation and $\rho :B_{n}\to {\left\{0,1\right\}}^{L-n}$ is a random function. Moreover, all these primitives are independently sampled. The PRF-based HCTR scheme (see Figure 6), denoted as HCTR^∗^, is defined below, which takes $\left(t,p∥p^{\prime }\right)$ as an input, and returns $c∥c^{\prime }$ as an output, where $p,c\in B_{n}$, t∈T and $|p^{\prime }\left|=\right|c^{\prime }|\leq L-n$. We call *t* the tweak, $p∥p^{\prime }$ the plaintext, and $c∥c^{\prime }$ the ciphertext.
–Input: t∈T, $p∥p^{\prime }$;Output: $c∥c^{\prime }$.
$x:=p⊕H(t,p^{\prime })$.y:=π(x).z:=x⊕y.$c^{\prime }:={\left\lfloor \rho \left(z\right)\right\rfloor }_{|p^{\prime }|}⊕p^{\prime }$.$c:=y⊕H(t,c^{\prime })$.

Note that the decryption algorithm is exactly same, except that we replace π by $\pi^{-1}$ in line (b). When we substitute ρ by ${\mathrm{CTR}}_{\pi }$, as in the standard encryption schemesusing IV-based counter mode [109], we obtain our original HCTR mode. The construction HCH [21] can be obtained by replacing ρ by the construction ${\mathrm{CTR}}_{\pi }$, where IV is computed as the encryption of *z* by π. We note that the original security bound for HCTR was cubic (in the number of queries). To obtain a quadratic bound, HCH was proposed. However, subsequently, in [108] a quadratic bound for HCTR was proven using the game playing technique. For the sake of simplicity, we provide a very simple proof of the HCTR* construction. The original HCTR scheme and all of its variants can be proven similarly.

**Lemma** **9.** 

$${\mathrm{Adv}}_{\mathrm{HCTR}*}^{\mathrm{sprp}}\left(q,\sigma \right)\leq q^{2}\cdot \left(\epsilon +2^{-n}\right).$$



**Proof.** (The basic idea of the proof is quite simple. We have to bound two types of collisions, namely, the input and output collisions on the underlying random permutation and the input collision of the random function. Most of these collisions can be bounded using the AXU property of H.)Extended System. Let F be the response system corresponding to the real system HCTRf and Π be the system corresponding to a length-preserving tweakable random permutation over the set of all bit strings of at least n+1 in size. The transcript random variable τ is defined as the tuple $\left(T^{q},P^{q},C^{q}\right)$, where for all i∈[q], $P_{i}$ and $C_{i}$ are of length $\ell_{i}$ and $\sum_{i\in [q]}\ell_{i}=\sigma$. We define the H-extended random system. In the real world, we simply release the hash key H after all *q* queries are made. Let ${\bar{F}}^{\pm }=\left(F^{\pm },H\right)$ be the extended real system. In the ideal system, we adjoin a dummy hash key $H\leftarrow_{\$}H$, chosen independently of Π. Let ${\bar{\Pi }}^{\pm }=\left(\Pi^{\pm },H\right)$ be the extended ideal system. We define the internal variables $X_{i}:=H\left(T_{i},P_{i}\right)$, and $Y_{i}:=H\left(T_{i},C_{i}\right)$, and $Z_{i}:=X_{i}⊕Y_{i}$.Analysis of Bad Transcripts. We say that a transcript is bad if one of the following conditions is met:
xcoll:∃i≠j∈[q], such that $X_{i}=X_{j}$.ycoll:∃i≠j∈[q], such that $Y_{i}=Y_{j}$.zcoll:∃i≠j∈[q], such that $Z_{i}=Z_{j}$.Bound on Pr[xcoll]: Note that H is sampled independently of the inputs $\left(T_{i},P_{i}\right)$. So, for any i≠j, $\mathrm{Pr}(X_{i}=X_{j})\leq \epsilon$ as H is the ϵ-AXU hash function. A similar bound works for $Y_{i}=Y_{j}$.Bound on Pr[ycoll]: Exactly the same bound works for $Y_{i}=Y_{j}$.Bound on Pr[zcoll]: Fix any i<j and let us assume that the *j*th query is the encryption query (a similar argument would work for the decryption query). Now, $Z_{i}=Z_{j}$ means that
$$C_{j}^{\prime }=Z_{i}⊕P_{j}^{\prime }⊕\left(H\left(T_{j},P_{j}\right)⊕H\left(T_{j},C_{j}\right)\right).$$
Note that $C_{j}^{\prime }$ is uniform and independent of all random variables present on the right-hand side of the above equation. Hence, the probability of the above event is $2^{-n}$.By summing over all pairs (i,j) with i<j, the probability that the ideal-world extended transcript is bad is, at most, $q^{2}\cdot \left(\epsilon +2^{-n-1}\right)$.Analysis of Good Transcripts. Fix a good transcript $\left(t^{q},p^{q},c^{q},h\right)$. Let q(ℓ,t) denote the number of queries of length *ℓ* for all ℓ∈[L] with the tweak *t*. By definition, $\sum_{\ell ,t}q\left(\ell ,t\right)=\sigma$. Since for a good transcript there are no input/output collisions for π. So we have
$$\begin{array}{cc} \mathrm{Pr}[F\left(t^{q},p^{q}\right)=c^{q},H=h] & =\frac{1}{\left|H\right|}\times \frac{1}{{\left(2^{n}\right)}_{q}}\times \frac{1}{2^{\sigma -nq}}\\ \; & \geq \frac{1}{\left|H\right|}\times \left(1-\frac{q^{2}}{2^{n+1}}\right)\times \frac{1}{2^{\sigma }}\\ \; & \geq \left(1-\frac{q^{2}}{2^{n+1}}\right)\times \frac{1}{\left|H\right|}\times \prod_{\ell ,t}\frac{1}{{\left(2^{\ell }\right)}_{q(\ell ,t)}}\\ \; & \geq \left(1-\frac{q^{2}}{2^{n+1}}\right)\times \mathrm{Pr}\left[\Pi \left(t^{q},p^{q}\right)=c^{q},H=h\right] \end{array}$$
The result follows from the extended H-technique. □

Now let us consider the original HCTR scheme, in which $c^{\prime }=m^{\prime }⊕{\left\lfloor S\right\rfloor }_{|m^{\prime }|}$, where S=π(z⊕1)∥⋯∥π(z⊕(ℓ−1)). Let us assume that *i*th queries have size $n\ell_{i}$, $\ell_{i}\leq \ell$. We need to consider a revised definition of a zcoll bad event. We say that zcoll holds if one of the following holds:
$Z_{i}⊕j=Z_{i^{\prime }}⊕j^{\prime }$ or $m_{i}^{\prime }\left[j\right]⊕c_{i}^{\prime }\left[j\right]=m_{i^{\prime }}^{\prime }\left[j^{\prime }\right]⊕c_{i^{\prime }}^{\prime }\left[j^{\prime }\right]$ for some $\left(i,j\right)\neq \left(i^{\prime },j^{\prime }\right)$, $j<\ell_{i}$, $j^{\prime }<\ell_{i^{\prime }}$,$Z_{i}⊕j=X_{k}$ or $m_{i}\left[j\right]⊕c_{i}\left[j\right]=Y_{k}$, $j<\ell_{i}$, k∈[q].

Using the previous analysis, one can easily verify that the probability of this modified zcoll is at most $\left(\sigma^{2}+\sigma q\right)\left(2\epsilon +2^{-n-3}\right)$ (as there are, at most, $\sigma^{2}$ choices for (i,j) and $\left(i^{\prime },j^{\prime }\right)$, and at most σq choices for (i,j) and *k*).

### 7.2. TET

Construction.TET (a later, simplified version was renamed HEH in [24]) is an encryption scheme developed by Halevi [20], based on the hash-encrypt-hash paradigm [20,24,25], which uses a sandwich consisting of the ECB mode in between two blockwise universal and invertible length-preserving hash functions. In this paper, we formally describe and analyze the hash-encrypt-hash paradigm defined over multiple messages of *n*-bits.

A family of hash functions $H:B_{n}^{\leq L}\overset{*}{\to }B_{n}^{\leq L}$ is called an *$\left(\epsilon_{1},\epsilon_{2}\right)$-blockwise universal* if it is length-preserving. For all l≤L, for all $m\in B_{n}^{l}$, $H\left(m\right)\in B_{n}^{l}$. and for all $\ell ,\ell^{\prime }\in \left[L\right]$, $m\in B_{n}^{\ell }$, $m^{\prime }\in B_{n}^{\ell^{\prime }}$, i∈[ℓ], and $i^{\prime }\in \left[\ell^{\prime }\right]$ with $\left(m,i\right)\neq \left(m^{\prime },i^{\prime }\right)$,
$${\mathrm{Pr}}_{H\leftarrow_{\$}H}\left[H\left(m\right)\left[i\right]=H\left(m^{\prime }\right)\left[i^{\prime }\right]\right]\leq \left\{\begin{array}{cc} \epsilon_{1} & \mathrm{if}\;i=i^{\prime },\\ \epsilon_{2} & \mathrm{otherwise}. \end{array}\right.$$
Suppose that H is an $\left(\epsilon_{1},\epsilon_{2}\right)$-blockwise universal and invertible hash function over $B_{n}^{\leq L}$. Suppose that π is an *n*-bit random permutation independent of the hash. Then, the composition $H^{-1}\circ {\mathrm{ECB}}_{\pi }\circ H$ is called the TET construction (see Figure 7), which is defined over $B_{n}^{\leq L}$. A trick such as HCTR or the DE (the domain expander reported by Nandi in [110]) can help to process arbitrary bit strings.

**Lemma** **10.** 

$${\mathrm{Adv}}_{\mathrm{TET}}^{\mathrm{sprp}}\left(q,L,\sigma \right)\leq q^{2}L\epsilon_{1}+\sigma^{2}\epsilon_{2}.$$



**Proof.** We will again use the same idea of avoiding collisions among the input/output of the internal random permutation π. Let F be the response system corresponding to TET and Π be the system corresponding to a random permutation. The transcript τ is defined as the tuple $\left(P^{q},C^{q}\right)$, where for all i∈q, $P_{i}$ and $C_{i}$ are of length $\ell_{i}$ and $\sum_{i\in [q]}\ell_{i}=\sigma$.Analysis of Good Transcripts. This proof will be similar to that of hash-then-PRF. For any transcript $\left(p^{q},c^{q}\right)$, we have
(21)$$\begin{array}{cc} \mathrm{Pr}[F\left(p^{q}\right)=c^{q}] & \geq \sum_{x^{q},y^{q}\in B_{n}^{\left(\sigma \right)}}\mathrm{Pr}\left[H\left(p^{q}\right)=x^{q},\pi \left(x^{q}\right)=y^{q},H^{-1}\left(y^{q}\right)=c^{q}\right]\\ \; & =\sum_{x^{q},y^{q}\in B_{n}^{\left(\sigma \right)}}\mathrm{Pr}\left[H\left(p^{q}\right)=x^{q},H\left(c^{q}\right)=y^{q}\right]\times \mathrm{Pr}\left[\pi \left(x^{q}\right)=y^{q}\right]\\ \; & =\mathrm{Pr}\left[H\left(p^{q}\right)\in B_{n}^{\left(\sigma \right)},\;H\left(c^{q}\right)\in B_{n}^{\left(\sigma \right)}\right]\times \frac{1}{{\left(N\right)}_{\sigma }}\\ \; & \geq \left(1-q^{2}L\epsilon_{1}-\sigma^{2}\epsilon_{2}\right)\times \mathrm{Pr}\left[\Pi \left(p^{q}\right)=c^{q}\right] \end{array}$$
The latter inequality follows from the definition of the blockwise universal hash function and from the observation $\mathrm{Pr}\left[\Pi \left(p^{q}\right)=c^{q}\right]\leq \frac{1}{{\left(N\right)}_{\sigma }}$. The result follows from substituting Equation (Equation 21) in the standard H-technique. □

**Remark** **3.** 
*We note that the bound in Lemma 10 is obtained for a slightly generalized definition of blockwise universality. Specifically, we consider two different bounds, one for collisions at the same position, and another for collisions at two different positions. This slight generalization gives a better security bound for certain hash functions that have better bounds for collisions at two different positions. For example, consider the following example from the work of Sarkar [24].*

*We view*

$B_{n}$

*as the finite field*

$\mathrm{GF}(2^{n})$

*and fix α to be a primitive element of*

$B_{n}$

*. Let*

$K_{1}$

*and*

$K_{2}$

*be two independent and random elements of*

$B_{n}$

*. Define*

$e_{\ell }=\left(\alpha K_{1},\alpha^{2}K_{1},\ldots ,\alpha^{\ell -1}K_{1},K_{1}\right)$

*, for all*

ℓ∈[L]

*. We define the map*

$H_{K_{1},K_{2}}:B_{n}^{\leq L}\to B_{n}^{\leq L}$

*in the following manner:*

$$H_{K_{1},K_{2}}\left(x\left[1\right],\ldots ,x\left[\ell \right]\right)=\left(x\left[1\right]⊕y,\ldots ,x\left[\ell -1\right]⊕y,y\right)⊕e_{\ell },$$

*where*

$y=\sum_{i=1}^{\ell }x\left[i\right]K_{2}^{\ell -i}$

*. Sarkar proved that H is an*

$\left(L2^{-n},2^{-n}\right)$

*-blockwise universal hash function ([24], Theorem 1). Using this bound in combination with Lemma 10 gives a bound of the form*

$\sigma^{2}2^{-n}+q^{2}L^{2}2^{-n}$

*, which results in a birthday bound in terms of L.*


## 8. Beyond-Birthday-Bound Secure Schemes

In this section, we revisit some beyond-birthday-bound secure schemes. All these schemes are inherently based on one of the most celebrated problems in symmetric-key cryptography, called the sum of permutations problem [59,96,97,99].

### 8.1. Pseudorandom Functions

There are many beyond-birthday-bound PRF constructions from PRP [61,96]. The sum of permutations [96,97,98] is one such construction, which constructs a PRF based on two independently keyed random permutations.

We first state and prove a simple proposition that lower-bounds the probability distribution of the sum of permutations conditioned on the event that the underlying random permutations are already sampled on some fixed number of points. We remark that a similar result is already available in ([64], Theorem 2), albeit for the single-permutation setting. Their analysis similarly can be extended for the two-permutation setting. For the sake of completeness, we provide the proof of this variant.

**Proposition** **2.** 
*Let*

$s_{1},s_{2},q\geq 0$

*and*

$s_{1}+s_{2}+2q\leq 2^{n-1}$

*. Let*

$\pi_{1}$

*and*

$\pi_{2}$

*be two independent and uniform random permutations over*

${\left\{0,1\right\}}^{n}$

*. For all*

$a^{s_{1}},b^{s_{1}}\in B_{n}^{\left(s_{1}\right)}$

*,*

$c^{s_{2}},d^{s_{2}}\in B_{n}^{\left(s_{2}\right)}$

*,*

$x^{q}\in {\left(B_{n}\setminus \left\{a^{s_{1}}\right\}\right)}^{\left(q\right)}$

*,*

$y^{q}\in {\left(B_{n}\setminus \left\{b^{s_{2}}\right\}\right)}^{\left(q\right)}$

*, and*

$z^{q}\in B_{n}^{q}$

*, we have*

$$\mathrm{Pr}\left[\pi_{1}\left(x^{q}\right)⊕\pi_{2}\left(y^{q}\right)=z^{q}\;|\;F\right]\geq \left(1-\frac{2s_{1}s_{2}q}{2^{2n}}-\frac{\left(s_{1}+s_{2}\right)q^{2}}{2^{2n}}-\frac{q^{3}}{2^{2n}}\right)\times \frac{1}{2^{nq}},$$

*where*

F

*denotes the event*

$\pi_{1}\left(a^{s_{1}}\right)=b^{s_{1}}\wedge \pi_{2}\left(c^{s_{2}}\right)=d^{s_{2}}$

*.*


**Proof.** We compute a lower bound on the probability by following the chain rule of conditional probabilities, i.e., we compute the conditional probability of $\pi_{1}\left(x_{i}\right)⊕\pi_{2}\left(y_{i}\right)=z_{i}$ for some i∈[q], given F, and $\pi_{1}\left(x_{j}\right)⊕\pi_{2}\left(y_{j}\right)=z_{j}$ for all j<i.Let $P_{i}=\pi_{1}\left(x_{i}\right)$, $Q_{i}=\pi_{2}\left(y_{i}\right)$, and $E_{i}$ denote the event that the *i*-th equation $P_{i}⊕Q_{i}=z_{i}$ holds, for all i∈[q].For all 1≤i≤q, consider the *i*-th equation $P_{i}⊕Q_{i}=z_{i}$. We have at least $2^{n}-s_{1}-s_{2}-2\left(i-1\right)$ possibilities for $P_{i}$. This can be argued by removing $b^{s_{1}}$, $d^{s_{2}}⊕z_{i}$, $P_{j}$ and $Q_{j}⊕z_{i}$ values for all j<i. Once we fix $P_{i}$, $Q_{i}$ fixes to $P_{i}⊕z_{i}$. Furthermore, for each such value, $E_{i}$ occurs with a probability of $1/\left(2^{n}-s_{1}-i+1\right)\left(2^{n}-s_{2}-i+1\right)$ given that F occurs, and $E_{j}$ occurs for all j<i. Let k=i−1. Then, we have
$$\begin{array}{cc} \mathrm{Pr}[E_{i}\;|\;F\wedge E_{1}\wedge \cdots \wedge E_{k}] & \geq \frac{2^{n}-s_{1}-s_{2}-2k}{\left(2^{n}-s_{1}-k\right)\left(2^{n}-s_{2}-k\right)}\\ \; & \geq \frac{2^{2n}-\left(s_{1}+s_{2}+2k\right)2^{n}}{2^{2n}-\left(s_{1}+s_{2}+2k\right)2^{n}+s_{1}s_{2}+\left(s_{1}+s_{2}\right)k+k^{2}}\times \frac{1}{2^{n}}\\ \; & =\left(1-\frac{s_{1}s_{2}+\left(s_{1}+s_{2}\right)k+k^{2}}{2^{2n}-\left(s_{1}+s_{2}+2k\right)2^{n}+s_{1}s_{2}+\left(s_{1}+s_{2}\right)k+k^{2}}\right)\times \frac{1}{2^{n}}\\ \; & \geq \left(1-\frac{2(s_{1}s_{2}+\left(s_{1}+s_{2}\right)k+k^{2})}{2^{2n}}\right)\times \frac{1}{2^{n}}, \end{array}$$
where the latter inequality follows from the assumptions that $\left(s_{1}+s_{2}+2k\right)2^{n}<\left(s_{1}+s_{2}+2q\right)2^{n}<2^{2n-1}$ and $s_{1}s_{2}+\left(s_{1}+s_{2}\right)k+k^{2}\geq 0$. Finally, we have
$$\begin{array}{cc} \mathrm{Pr}[\pi_{1}\left(x^{q}\right)⊕\pi_{2}\left(y^{q}\right)=z^{q}\;|\;F] & =\mathrm{Pr}\left[E_{1}\;|\;F\right]\times \mathrm{Pr}\left[E_{2}\;|\;F\wedge E_{1}\right]\times \cdots \times \mathrm{Pr}\left[E_{q}\;|\;F\wedge E_{1}\wedge \cdots \wedge E_{q-1}\right]\\ \; & \geq \prod_{k=0}^{q-1}\left(1-\frac{2(s_{1}s_{2}+\left(s_{1}+s_{2}\right)k+k^{2})}{2^{2n}}\right)\times \frac{1}{2^{nq}}\\ \; & \geq \left(1-2\sum_{k=0}^{q-1}\frac{s_{1}s_{2}+\left(s_{1}+s_{2}\right)k+k^{2}}{2^{2n}}\right)\times \frac{1}{2^{nq}}\\ \; & \geq \left(1-\frac{2s_{1}s_{2}q}{2^{2n}}-\frac{\left(s_{1}+s_{2}\right)q^{2}}{2^{2n}}-\frac{q^{3}}{2^{2n}}\right)\times \frac{1}{2^{nq}}. \end{array}$$□

Note that Proposition 2 is useful in both conditional and unconditional (i.e., $s_{1}=s_{2}=0$) cases. Now, we discuss two straightforward applications of the proposition given above.

#### 8.1.1. Sum of Permutations

Construction: Suppose that $\pi_{1}$ and $\pi_{2}$ are two independent uniform random permutations over ${\left\{0,1\right\}}^{n}$. The sum of permutation (or SoP) construction [96,97,98] (illustrated in Figure 8) is a length-preserving function over ${\left\{0,1\right\}}^{n}$, defined by the mapping
$$x\overset{\mathrm{SoP}}{⟼}\pi_{1}\left(x\right)⊕\pi_{2}\left(x\right).$$

It is well-known [89,111,112] that SoP is indistinguishable from a uniform random function up to $o(2^{n})$ queries. The H-technique-based proofs presented in [111,112], although tight, contain some non-trivial gaps, whereas the proof in [89] uses the recently introduced $\chi^{2}$ technique.

In one of the early works on this problem, Lucks [98] presented a suboptimal bound of $2^{2n/3}$ queries using the game-playing technique. Here, we give a very simple and short proof for the $2^{2n/3}$ query bound using the standard H-technique.

**Lemma** **11.** 
*For*

$q\leq 2^{2n/3}$

*,*

$${\mathrm{Adv}}_{{\mathrm{SoP}}_{\pi_{1},\pi_{2}}}^{\mathrm{prf}}\left(q\right)\leq \frac{q^{3}}{2^{2n}}.$$



**Proof.** We write $F={\mathrm{SoP}}_{\pi_{1},\pi_{2}}$. Let ρ be a uniform random function over ${\left\{0,1\right\}}^{n}$. Let $\tau =(x^{q},y^{q})$ denote a transcript where $x^{q}\in {\left({\left\{0,1\right\}}^{n}\right)}^{\left(q\right)}$. All transcripts are considered to be good.Analysis of Good Transcripts. Given a transcript $\tau =(x^{q},y^{q})$, it is easy to see that
(22)$$\mathrm{Pr}\left[\rho \left(x^{q}\right)=y^{q}\right]=\frac{1}{2^{nq}},$$
as $x^{q}⇝y^{q}$. For the lower bound on $\mathrm{Pr}[F\left(x^{q}\right)=y^{q}]$, we summon Proposition 2, with $s_{1}=0$ and $s_{2}=0$, i.e., the random permutations are not sampled at any points as of now. Accordingly, we have
$$\begin{array}{cc} \mathrm{Pr}[F\left(x^{q}\right)=y^{q}] & =\left(1-\frac{q^{3}}{2^{2n}}\right)\times \frac{1}{2^{nq}}\\ \; & \geq \left(1-\frac{q^{3}}{2^{2n}}\right)\times \mathrm{Pr}\left[\rho \left(x^{q}\right)=y^{q}\right]\\ \frac{\mathrm{Pr}[F\left(x^{q}\right)=y^{q}]}{\mathrm{Pr}[\rho \left(x^{q}\right)=y^{q}]} & \geq \left(1-\frac{q^{3}}{2^{2n}}\right). \end{array}$$
The result follows from the standard H-technique. □

#### 8.1.2. Sum of Even-Mansour

In a recent paper [99], Chen et al. presented various beyond-birthday-bound secure PRF constructions based on public (i.e., in which the adversary has oracle access to the underlying random permutations) random permutations. Here we consider SoEM22, or the sum of even-Mansour scheme, with two independent random permutations and two independent keys.

Construction: Suppose that $\pi_{1}$ and $\pi_{2}$ are two independent uniform random permutations over ${\left\{0,1\right\}}^{n}$ and $\left(K_{1},K_{2}\right)\leftarrow_{\$}{\left\{0,1\right\}}^{2n}$. The SoEM22 construction (illustrated in Figure 9) is a length-preserving function over ${\left\{0,1\right\}}^{n}$ defined by the mapping
$$x\overset{\mathrm{SoEM}22}{\to }\pi_{1}\left(x⊕K_{1}\right)⊕K_{1}⊕\pi_{2}\left(x⊕K_{2}\right)⊕K_{2}.$$

In [99], SoEM22 has been shown to be a secure PRF up to $o(2^{2n/3})$ queries to both the underlying permutations, as well as the construction itself. Furthermore, it has been shown that the bound is tight. We demonstrate a similar security bound using Proposition 2.

**Lemma** **12.** 
*Let q and p denote the total number of constructions and primitive queries, respectively. Then, for*

$q,p\leq 2^{2n/3}$

*, we have*

$${\mathrm{Adv}}_{\mathrm{SoEM}22}^{\mathrm{prf}}\left(q,p\right)\leq \frac{2p^{2}q}{2^{2n}}+\frac{9pq^{2}}{2^{2n}}+\frac{5q^{3}}{2^{2n}}.$$



**Proof.** We can write $F=\mathrm{SoEM}22_{\pi_{1},\pi_{2},K_{1},K_{2}}$. Let ρ be a uniform random function over ${\left\{0,1\right\}}^{n}$. Let $\tau_{C}=\left(m^{q},z^{q}\right)$ denote the transcript corresponding to F, where $x^{q}\in {\left({\left\{0,1\right\}}^{n}\right)}^{\left(q\right)}$. Without the loss of generality, we assume that the adversary makes the same number of queries to the underlying random permutations. Let $\tau_{P}=\left(u^{p},v^{p}\right)$ and $\left(x^{p},y^{p}\right)$ denote the forward-only transcript corresponding to the direct access of $\pi_{1}^{\pm }$ and $\pi_{2}^{\pm }$, respectively.We define ${\left\{0,1\right\}}^{2n}$-extended response systems by adjoining the masking keys $K_{1}$ and $K_{2}$. In the case of F, this is well-defined according to the definition of SoEM22. In the ideal system we sample $\left(K_{1},K_{2}\right)\leftarrow_{\$}{\left\{0,1\right\}}^{2n}$. Note that, once $K_{1}$ and $K_{2}$ are released, one can easily obtain $A^{q}$, $C^{q}$, $B^{q}$, and $D^{q}$.Bad Transcript and Its Analysis. A transcript is called bad precisely when it leads to permutation incompatibility for any one of the underlying permutations. One way to avoid such inconsistencies is to avoid input/output collision constraints over the two permutations simultaneously. More formally, we say that a transcript is bad if one of the following events occurs for some $\left(M_{i},Z_{i}\right)\in \phi_{c},\left(U_{j},V_{j}\right)\in \phi_{1},\left(X_{j^{\prime }},Y_{j^{\prime }}\right)\in \phi_{2}$,
B1: $A_{i}=U_{j}$ and $C_{i}=X_{j^{\prime }}$.B2: $A_{i}=U_{j}$ and $B_{i}⊕Z_{i}⊕K_{1}⊕K_{2}=Y_{j^{\prime }}$.B3: $C_{i}=X_{j^{\prime }}$ and $D_{i}⊕Z_{i}⊕K_{1}⊕K_{2}=V_{j}$.Here, B1 corresponds to a collision on the input of the two underlying permutations; B2 corresponds to the input collision on $\pi_{1}$ and the output collision on $\pi_{2}$; and, B3 corresponds to the input collision on $\pi_{2}$ and the output collision on $\pi_{1}$. Note that we only consider collisions between constructions and primitive queries. This is because of the fact that construction-to-construction and primitive-to-primitive collisions are forbidden by design. Now, we have
$$\begin{array}{cc} \mathrm{Pr}[\bar{\tau }\left(A^{\rho ,\pi_{1}^{\pm },\pi_{2}^{\pm }}\right)\in \Omega_{\mathrm{bad}}] & \leq \sum_{i,j,j^{\prime }}\mathrm{Pr}\left[B1\right]+\mathrm{Pr}\left[B2\right]+\mathrm{Pr}\left[B3\right]\leq \frac{3qp^{2}}{2^{2n}}, \end{array}$$
where the latter inequality can be argued based on the fact that there are, at most, $qp^{2}$$\left(i,j,j^{\prime }\right)$ triples, and for each such triple, $\mathrm{Pr}\left[\mathrm{Bi}\right]=1/2^{2n}$ for all i∈[3]. This is because each bad event reduces to a system of two linear equations in two independent and uniform random variables $K_{1}$ and $K_{2}$; therefore, a solution occurs with $1/2^{2n}$ probability.Analysis of Good Transcripts. Given a transcript $\tau =(x^{q},y^{q})$, it is easy to see that
(23)$$\mathrm{Pr}\left[\rho \left(x^{q}\right)=y^{q},\pi_{1}\left(u^{p}\right)=v^{p},\pi_{2}\left(x^{p}\right)=y^{p},K_{1},K_{2}\right]=\frac{1}{2^{nq}}\times \frac{1}{{\left(2^{n}\right)}_{p}}\times \frac{1}{{\left(2^{n}\right)}_{p}}\times \frac{1}{2^{2n}},$$
as $m^{q}⇝z^{q}$, $u^{p}↭v^{p}$, and $x^{p}↭y^{p}$ and $\left(K_{1},K_{2}\right)$ are chosen uniformly from ${\left\{0,1\right\}}^{2n}$. In the real world, for primitive queries we know that $u^{p}↭v^{p}$, and $x^{p}↭y^{p}$. In construction queries, the *i*-th query could be one of the three types: (1) $a_{i}=u_{j}$ and $c_{i}\neq x_{j^{\prime }}$ for all $j^{\prime }\in \left[p\right]$; (2) $a_{i}\neq u_{j}$ for all j∈[p] and $c_{i}=x_{j^{\prime }}$; (3) $a_{i}\neq u_{j}$ and $c_{i}\neq x_{j^{\prime }}$ for all $j,j^{\prime }\in \left[p\right]$. It is easy to see that both *a* and *c* cannot collide simultaneously as the transcript is good. Let $\phi_{C}^{1}$, $\phi_{C}^{2}$, and $\phi_{C}^{3}$ denote type-1, type-2, and type-3 construction transcripts, respectively, and $q_{1}$, $q_{2}$, and $q_{3}$ denote the number of such queries, respectively. Finally, we have
$$\begin{array}{cc} \mathrm{Pr}[F\left(x^{q}\right)=y^{q},\pi_{1}\left(u^{p}\right)=v^{p},\pi_{2}\left(x^{p}\right)=y^{p},K_{1},K_{2}] & =\mathrm{Pr}[\phi_{C},\phi_{P},K_{1},K_{2}]\\ \; & =\mathrm{Pr}\left[\phi_{P}\right]\times \mathrm{Pr}\left[\phi_{C}^{1}\;|\;\phi_{P}\right]\times \mathrm{Pr}\left[\phi_{C}^{2}\;|\;\phi_{P}\wedge \phi_{C}^{1}\right]\\ \; & \;\;\;\;\times \mathrm{Pr}\left[\phi_{C}^{3}\;|\;\phi_{P}\wedge \phi_{C}^{1}\wedge \phi_{C}^{2}\right]\times \frac{1}{2^{2n}}\\ \; & =\frac{1}{2^{2n}}\times \frac{1}{{\left(2^{n}\right)}_{p}{\left(2^{n}\right)}_{p}}\times \frac{1}{{\left(2^{n}-p\right)}_{q_{1}}}\\ \; & \;\;\;\;\times \frac{1}{{\left(2^{n}-p\right)}_{q_{2}}}\times \mathrm{Pr}\left[\phi_{C}^{3}\;|\;\phi_{P}\wedge \phi_{C}^{1}\wedge \phi_{C}^{2}\right] \end{array}$$
Observe that we can apply Proposition 2 to bound the conditional probability of $\phi_{C}^{3}$ given $\phi_{P}$, $\phi_{C}^{1}$, and $\phi_{C}^{2}$. Thus, by using $s_{1}=p+q_{1}$, $s_{2}=p+q_{2}$, and the relation $q_{1},q_{2},q_{3}<q$, we have
(24)$$\begin{array}{cc} \mathrm{Pr}[F\left(x^{q}\right)=y^{q},\pi_{1}\left(u^{p}\right)=v^{p},\pi_{2}\left(x^{p}\right)=y^{p},K_{1},K_{2}] & \geq \left(1-\frac{2p^{2}q}{2^{2n}}-\frac{6pq^{2}}{2^{2n}}-\frac{5q^{3}}{2^{2n}}\right)\times \frac{1}{2^{nq_{3}}}\\ \; & \;\;\;\;\times \frac{1}{{\left(2^{n}\right)}_{p}{\left(2^{n}\right)}_{p}}\times \frac{1}{{\left(2^{n}-p\right)}_{q_{1}}}\\ \; & \;\;\;\;\times \frac{1}{{\left(2^{n}-p\right)}_{q_{2}}}\times \frac{1}{2^{2n}}. \end{array}$$
The result follows by dividing Equation (Equation 24) by Equation (Equation 23).□

## 9. Optimality of the Extended H-Technique

We have already observed that the expectation method can achieve optimal bounds for the distinguishing advantage. The extended H-technique is also a potential tool to obtain a tight bound for the distinguishing advantage. Now we describe why this is the case. Suppose that F and α are two (X,Y) random systems. We usually choose α to be an ideal random system (such as a random permutation or a random function) and F is the construction of interest. Let
$$\begin{array}{cc} E_{F\geq \alpha } & =\left\{\left(x^{q},y^{q}\right)\;|\;r^{F/\alpha }\left(x^{q},y^{q}\right):=\frac{\mathrm{Pr}[F\left(x^{q}\right)=y^{q}]}{\mathrm{Pr}[\alpha \left(x^{q}\right)=y^{q}]}\geq 1\right\}. \end{array}$$
The complement of the above set is denoted as $E_{F<\alpha }$. We also define a binary random variable B adjoined with α as follows. Let
(25)$$\begin{array}{cc} \mathrm{Pr}[B=0\;|\;\alpha \left(x^{q}\right)=y^{q}] & =1,\;\;\forall \left(x^{q},y^{q}\right)\in E_{F\geq \alpha } \end{array}$$
and
(26)$$\begin{array}{cc} \mathrm{Pr}[B=0\;|\;\alpha \left(x^{q}\right)=y^{q}] & =r^{F/\alpha }\left(x^{q},y^{q}\right),\;\;\forall \left(x^{q},y^{q}\right)\in E_{F<\alpha }. \end{array}$$
We can combine these two equations and write the following for all $\left(x^{q},y^{q}\right)$:
(27)$$\mathrm{Pr}\left[B=1\;|\;\alpha \left(x^{q}\right)=y^{q}\right]=\max\left\{0,1-r^{F/\alpha }\left(x^{q},y^{q}\right)\right\}$$
and hence
(28)$$\mathrm{Pr}\left[B=1,\alpha \left(x^{q}\right)=y^{q}\right]=\max\left\{0,\mathrm{Pr}\left[\alpha \left(x^{q}\right)=y^{q}\right]-\mathrm{Pr}\left[F\left(x^{q}\right)=y^{q}\right]\right\}.$$
We say that a transcript $\left(x^{q},y^{q},b\right)$ is bad if b=1. Fix any deterministic adversary A. The probability that the extended transcript random variable $\bar{\tau }\left(A^{\alpha }\right)$ is bad is
$$\begin{array}{cc} \mathrm{Pr}[B=1] & =\sum_{A\left(y^{q}\right)=x^{q}}\max\left\{0,\mathrm{Pr}\left[\alpha \left(x^{q}\right)=y^{q}\right]-\mathrm{Pr}\left[F\left(x^{q}\right)=y^{q}\right]\right\}\\ \; & =\Delta (\tau \left(A^{F}\right)\;;\;\tau \left(A^{\alpha }\right)). \end{array}$$
Now we define $B^{\prime }$ adjoined with F. The random variable $B^{\prime }$ is degenerated and takes the value of zero with a probability of one. In other words, $\mathrm{Pr}\left[F\left(x^{q}\right)=y^{q},B^{\prime }=0\right]=\mathrm{Pr}\left[F\left(x^{q}\right)=y^{q}\right]$. It is easy to see that for all $\left(x^{q},y^{q}\right)$,
$$\begin{array}{c} \mathrm{Pr}\left[F\left(x^{q}\right)=y^{q},B^{\prime }=0\right]\geq \mathrm{Pr}\left[\alpha \left(x^{q}\right)=y^{q},B=0\right]. \end{array}$$

Thus, if we apply the H-technique we actually obtain equality in Equation (Equation 7).

**Remark** **4.** 
*Here we remark that although the extended H-technique and the expectation method can achieve optimal distinguishing bounds, this might require a very involved analysis. For the expectation method, identifying the optimal ϵ function and then provided a tight estimation for the expectation of this function can be quite difficult. Similarly, for the extended H-technique, identifying the optimal bad event can be very difficult. One thing is clear, however. Both these tools can achieve optimality whenever it is possible through the game-playing or random systems methodologies.*


### Nonadaptive PRP to SPRP

“Two weak make one strong” or the composition lemma [75,76] states that, in the information-theoretic setting, the composition of two NPRP secure-block ciphers results in an SPRP secure-block cipher. The initial proofs [75,76] of this result were based on Maurer’s random systems methodology. Subsequently, Cogliati, Patarin and Seurin [113] presented a much simpler proof using the standard H-technique.

Construction. Let F and G be two NPRP secure quasi-random permutations over X. Then, we are interested in the SPRP security of the composition $G^{-1}\circ F$. Formally, the composition result is stated in Theorem 1.

**Theorem** **1.**
*Suppose that*

F

*and*

G

*are two random systems over*

X

*; then,*

$${\mathrm{Adv}}_{G^{-1}\circ F}^{\mathrm{sprp}}\left(q\right)\leq {\mathrm{Adv}}_{F}^{\mathrm{nprp}}\left(q\right)+{\mathrm{Adv}}_{G}^{\mathrm{nprp}}\left(q\right).$$



In [113], the following result has been proven. Lemma 13 gives a simple proof for Theorem 1 using the standard H-technique.

**Lemma** **13.**
*For all*

$x^{q},y^{q}\in X^{\left(q\right)}$

*, we have*

$$\frac{\mathrm{Pr}[G^{-1}\circ F\left(x^{q}\right)=y^{q}]}{\mathrm{Pr}[\pi \left(x^{q}\right)=y^{q}]}\geq 1-{\mathrm{Adv}}_{F}^{\mathrm{nprp}}\left(q\right)-{\mathrm{Adv}}_{G}^{\mathrm{nprp}}\left(q\right).$$



**Alternate proof using the extended-H technique.** We give a similar but alternative proof for Theorem 1, using the idea of optimality of the extended H-technique. Since we will employ the extended H-technique, we start off with a description of the extended systems.

EXTENDED SYSTEMS. We consider $\left(X^{q}\times {\left\{0,1\right\}}^{2}\right)$-extended random systems. We first define a triple of random variables $\left(Z^{q},B_{1},B_{2}\right)\leftarrow X^{q}\times \left\{0,1\right\}\times \left\{0,1\right\}$ adjoined with $\pi^{\pm }$. Let $\tau \left(A^{\pi^{\pm }}\right):=\left(\delta^{q},X^{q},Y^{q}\right)$ be the forward-only transcript. We sample $Z^{q}\overset{\mathrm{wor}}{⟵}X$ independently of π (and hence independently of the transcript τ as well). Now, we define the conditional distribution of $B_{1},B_{2}$ given $\left(\delta^{q},X^{q},Y^{q},Z^{q}\right)$.

Fix three tuples $x^{q},y^{q},z^{q}\in X^{\left(q\right)}$. We define the distributions of $B_{1},B_{2}$ given that $X^{q}=x^{q},Y^{q}=y^{q}$ and $Z^{q}=z^{q}$. Note that the sampling of $Z^{q}$ can be viewed as $\pi^{\prime }\left(X^{q}\right)$ for a random permutation $\pi^{\prime }$ independently of π. If $\left(x^{q},z^{q}\right)\in E_{F\geq \pi^{\prime }}$ then $B_{1}=0$ with a probability of one. Otherwise, $B_{1}$ follows the Bernoulli distribution $\mathrm{ber}(1-r^{F/\pi^{\prime }}\left(x^{q},z^{q}\right))$. Similarly, if $\left(y^{q},z^{q}\right)\in E_{G\geq \pi^{\prime }}$ then $B_{2}=0$ with a probability of one. Otherwise, $B_{2}$ follows the Bernoulli distribution $\mathrm{ber}(1-r^{G/\pi^{\prime }}\left(y^{q},z^{q}\right))$.

Analysis of bad transcripts. We say that a transcript $\left(x^{q},y^{q},z^{q},b_{1},b_{2}\right)$ is bad if $b_{1}=1\vee b_{2}=1$. For the random transcript variable $\left(X^{q},Y^{q},Z^{q},B_{1},B_{2}\right)$, we denote this event as bad. According to the union bound,
$$\mathrm{Pr}\left[\mathrm{bad}\right]\leq \mathrm{Pr}\left[B_{1}=1\right]+\mathrm{Pr}\left[B_{2}=1\right].$$
We now show that $\mathrm{Pr}\left[B_{1}=1\right]\leq {\mathrm{Adv}}_{F}^{\mathrm{nprp}}\left(q\right)$. Similarly, one can show that $\mathrm{Pr}\left[B_{2}=1\right]\leq {\mathrm{Adv}}_{G}^{\mathrm{nprp}}\left(q\right)$. Similarly to Equation (Equation 28), we have
$$\begin{array}{cc} \mathrm{Pr}[B_{1}=1] & =\sum_{\begin{array}{c} (x^{q},y^{q})\in A\\ z^{q}\in X^{\left(q\right)} \end{array}}\mathrm{Pr}\left[B_{1}=1,Z^{q}=z^{q},X^{q}=x^{q},Y^{q}=y^{q}\right]\\ \; & =\sum_{\begin{array}{c} (x^{q},y^{q})\in A\\ z^{q}\in X^{\left(q\right)} \end{array}}\mathrm{Pr}\left[B_{1}=1,\pi^{\prime }\left(x^{q}\right)=z^{q},\pi \left(x^{q}\right)=y^{q}\right]\\ \; & =\sum_{(x^{q},y^{q})\in A}\mathrm{Pr}\left[\pi \left(x^{q}\right)=y^{q}\right]\sum_{z^{q}\in X^{\left(q\right)}}\mathrm{Pr}\left[\pi^{\prime }\left(x^{q}\right)=z^{q}\right]\times \max\left\{0,1-r^{F/\pi^{\prime }}\left(x^{q},z^{q}\right)\right\}\\ \; & =\sum_{(x^{q},y^{q})\in A}\mathrm{Pr}\left[\pi \left(x^{q}\right)=y^{q}\right]\sum_{z^{q}\in X^{\left(q\right)}}\max\left\{0,\mathrm{Pr}\left[\pi^{\prime }\left(x^{q}\right)=z^{q}\right]-\mathrm{Pr}\left[F\left(x^{q}\right)-z^{q}\right]\right\}\\ \; & =\sum_{(x^{q},y^{q})\in A}\mathrm{Pr}\left[\pi \left(x^{q}\right)=y^{q}\right]\times ∥{\mathrm{Pr}}_{\pi^{\prime }\left(x^{q}\right)}-{\mathrm{Pr}}_{F(x^{q})}∥\\ \; & \leq \max_{a^{q}\in X^{\left(q\right)}}∥{\mathrm{Pr}}_{\pi^{\prime }\left(a^{q}\right)}-{\mathrm{Pr}}_{F(a^{q})}∥\sum_{(x^{q},y^{q})\in A}\mathrm{Pr}\left[\pi \left(x^{q}\right)=y^{q}\right]\\ \; & \leq {\mathrm{Adv}}_{F}^{\mathrm{nprp}}\left(q\right). \end{array}$$

Analysis of good transcripts. We define $B_{1}^{\prime }$ and $B_{2}^{\prime }$ adjoined with F in a similar fashion as in the case of the optimality result. Both $B_{1}^{\prime }$ and $B_{2}^{\prime }$ are degenerated and take a value of zero with a probability of one. For $x^{q},y^{q},z^{q}\in X^{\left(q\right)}$ and i∈{1,2}, let $p_{i}:=\mathrm{Pr}\left[B_{i}=0\;\right|\;\pi \left(x^{q}\right)=y^{q}$, $Z^{q}=z^{q}]$. Then, we have
$$\begin{array}{cc} \mathrm{Pr}[\pi \left(x^{q}\right)=y^{q},Z^{q}=z^{q},B_{1}=0,B_{2}=0] & =\mathrm{Pr}\left[\pi \left(x^{q}\right)=y^{q}\right]\times \mathrm{Pr}\left[Z^{q}=z^{q}\right]\times p_{1}\times p_{2}\\ \; & =\mathrm{Pr}\left[\pi \left(x^{q}\right)=y^{q}\right]\times \mathrm{Pr}\left[Z^{q}=z^{q}\right]\times p_{1}\times p_{2}\\ \; & \leq \mathrm{Pr}\left[F\left(x^{q}\right)=z^{q}\right]\times \mathrm{Pr}\left[G\left(y^{q}\right)=z^{q}\right]\\ \; & =\mathrm{Pr}[F\left(x^{q}\right)=z^{q},G\left(y^{q}\right)=z^{q},B_{1}^{\prime }=0,B_{2}^{\prime }=0]\\ \; & =\mathrm{Pr}[G^{-1}\circ F\left(x^{q}\right)=y^{q},B_{1}^{\prime }=0,B_{2}^{\prime }=0] \end{array}$$
The result follows from the extended H-technique, Lemma 3.

## 10. Conclusions

In this systematization of knowledge, our main goal was to revisit a popular tool in symmetric-key provable security, called the (Coefficients) H-technique [36,38]. We re-formalized the notations and conventions necessary to study the security of any symmetric-key design. We then described the H-technique tool and showed that it can achieve optimal security bounds. To illustrate the effectiveness of this tool, we presented simple security proofs for some popular symmetric-key designs, across different paradigms.

Although our main goal is to promote the application of the H-technique, we emphasize that it is not a universal solution. In particular, there are many problems in which a straightforward application of the H-technique may not provide a tight bound. A prime example is the sum of permutations or SoP problem [96,97]. Although there are some tight-bound proofs [111,112] for the SoP problem that use the H-technique, the veracity of these proofs is not yet established. In contrast, a recent tool developed by Dai et al., called the $\chi^{2}$-method [89], provides a much simple and asymptotically tight bound proof for SoP. The preceding example is just one such instance in which one tool is somewhat superior to another. There could be many more. For instance, it is still not clear how one can apply the $\chi^{2}$-method with the same ease as the H-technique to the analysis of schemes based on low-entropy primitives (such as universal hash functions).

To conclude, a thorough study on the various available tools is presently needed. This will help in choosing the right tools for a given set of problems, which in turn may present tight and/or simple proofs. We believe that this work is a step in that direction. It would be interesting to see similar work on some other popular tools such as the coupling technique [85,86,87] and $\chi^{2}$-method [89,90]. At the same time, we must also look at some other avenues in probability theory to obtain new tools. For example, Morris, Rogaway, and Stegers explored the applications of Markov chains [86] and Steinberger explored the Hellinger distance [94]. 

## Figures and Tables

**Figure 1 entropy-24-00462-f001:**
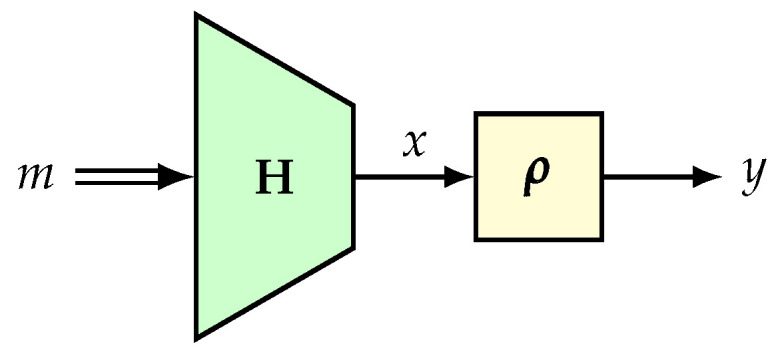
The hash-then-PRF paradigm. The double equal sign path denotes the possibility of a large message size.

**Figure 2 entropy-24-00462-f002:**
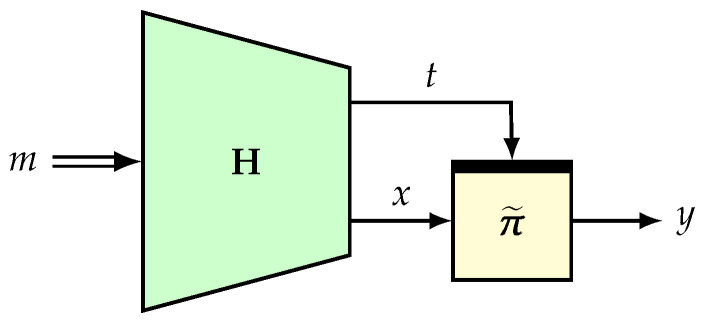
The hash-then-TPRP paradigm. The double equal sign path denotes the possibility of a large message size.

**Figure 3 entropy-24-00462-f003:**
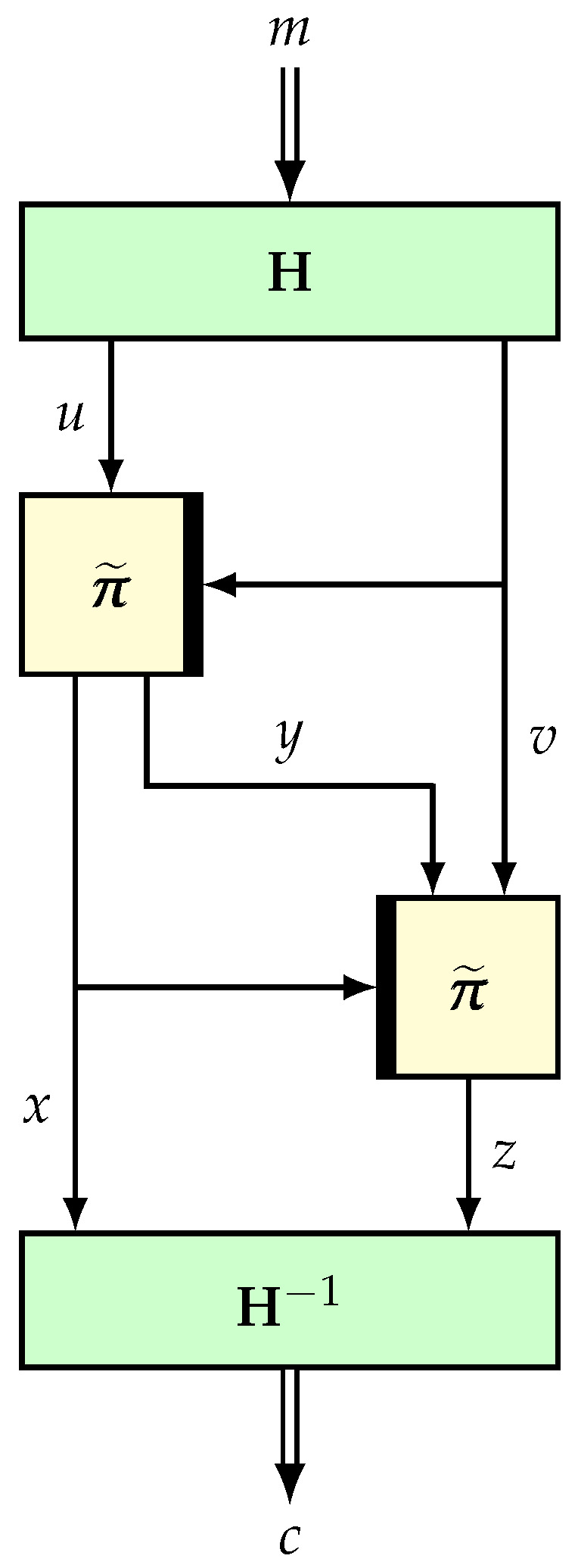
The NR* paradigm. The double equal sign path denotes an (n+t)-bit message and ciphertext.

**Figure 4 entropy-24-00462-f004:**
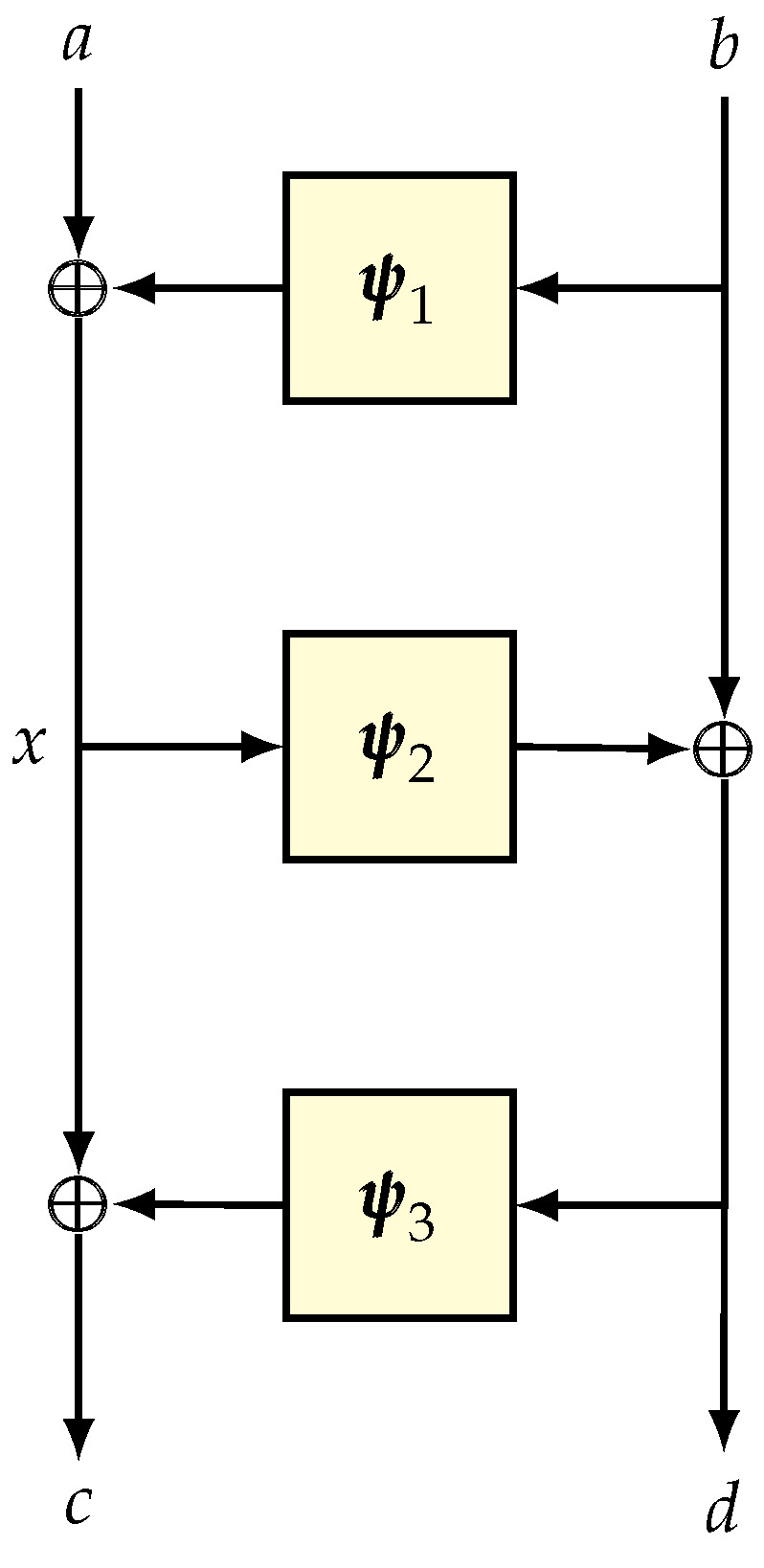
The three-round Luby–Rackoff or LR3 construction.

**Figure 5 entropy-24-00462-f005:**
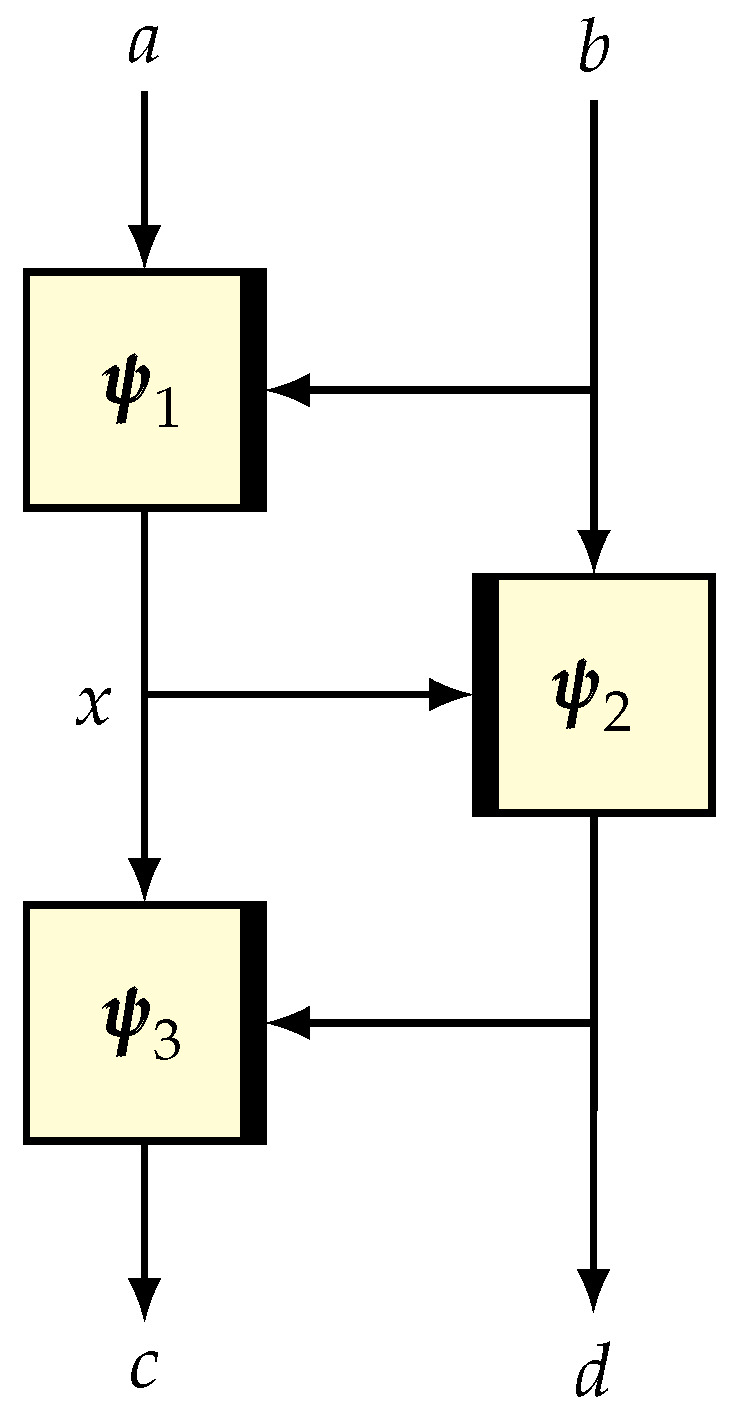
The three-round TPRP-based Luby–Rackoff or TLR3 construction.

**Figure 6 entropy-24-00462-f006:**
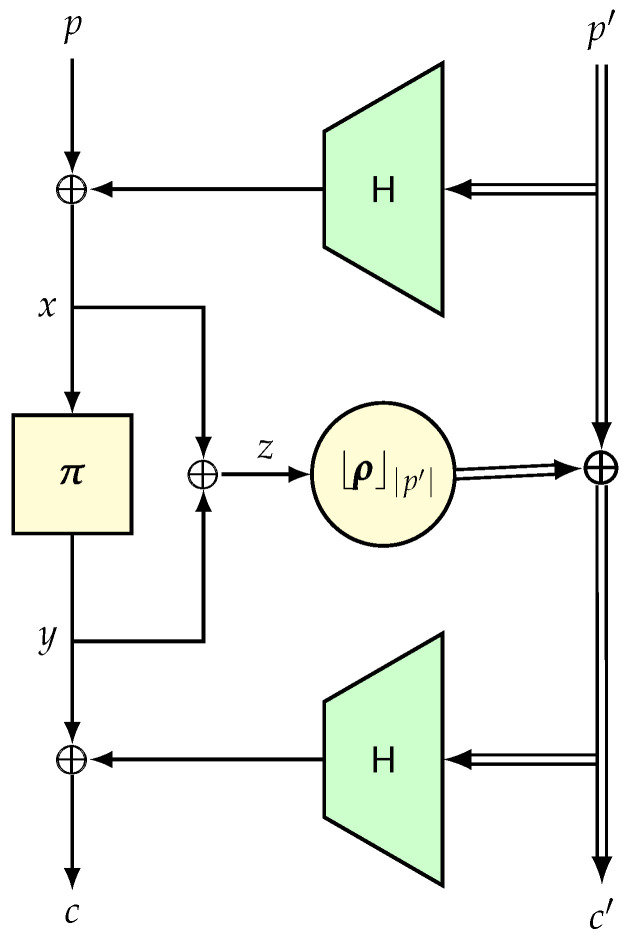
A simplified view of the HCTR enciphering scheme. The double equal style paths denote a compressed view of (ℓ−1) many parallel paths.

**Figure 7 entropy-24-00462-f007:**
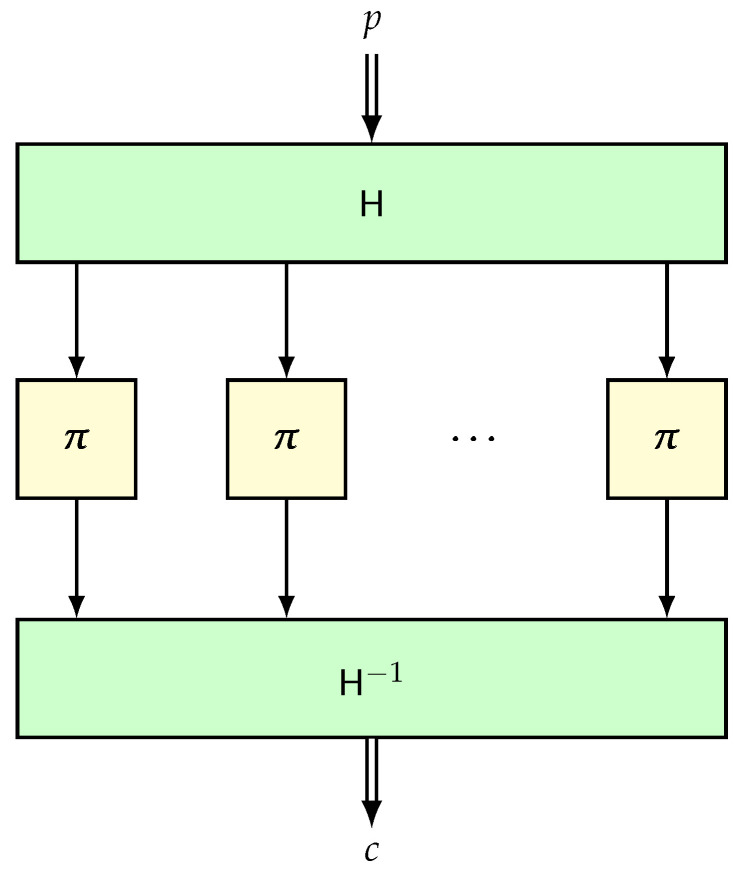
A simplified view of the TET enciphering scheme.

**Figure 8 entropy-24-00462-f008:**
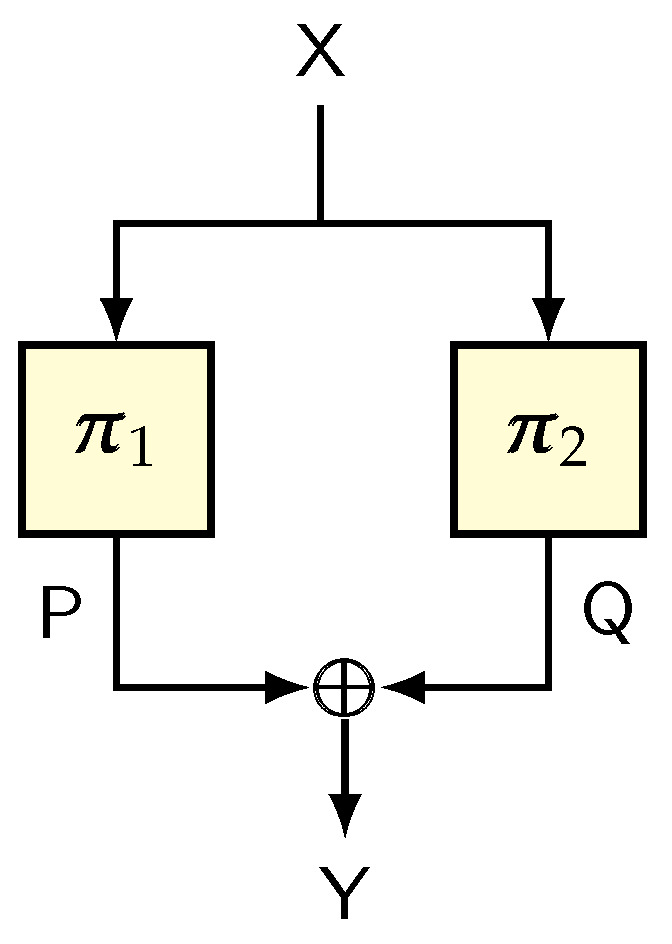
The sum of permutations construction.

**Figure 9 entropy-24-00462-f009:**
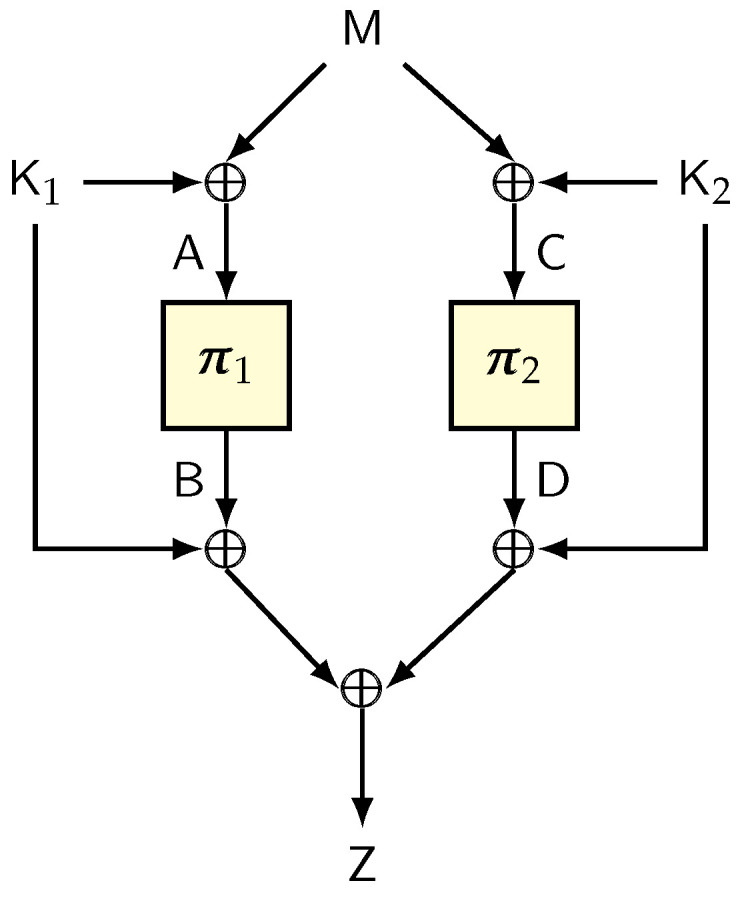
The sum of even-Mansour construction.

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
