# Peer review of "A Survey on Applications of H-Technique: Revisiting Security Analysis of PRP and PRF"

_entropy, 2022, doi:10.3390/e24040462_

Round 1
Reviewer 1 Report
The article is well written and easy to understand. However, few of my feedback can be considered to improve the quality of the paper but all are not necessary.
- Introduction may be improved, adding the highlights and the problem statements.
- You could improve writing, link better the ideas flow in the Introduction.
- Review references because some of them are unstandardized.
- The conclusion needs improvements towards major claimed contribution.
- Write some future directions in the conclusion section.
- The difference between your proposal and related works is not clear, you could to details better. I suggest add a comparative table in ''Related Literature'' to contrast your solution in front of related works. You have added too many references but because they are not in comparison form they seems boring to read.
- You could also discuss the relationship between your solution and past literature.
Author Response
1. Introduction may be improved, adding the highlights and the problem statements.
>> We think the current text in the introduction is adequately written. It first gives a brief description for all the popular proof techniques in symmetric-key cryptography and then in the contribution section clearly state that the goal is to formalize and systematize the H-coefficient technique with applications to several schemes.
2. You could improve writing, link better the ideas flow in the Introduction.
>> See response to 1 above.
3. Review references because some of them are unstandardized.
>> We could not find any discrepancies in the references. We request the reviewer to kindly clarify the unstandardized point.
4. The conclusion needs improvements towards major claimed contribution.
>> We have tried to give a very brief discussion on contributions, since most of the results are already known, and proven via a different proof technique other than H-technique.
5. Write some future directions in the conclusion section.
>> The last two paragraphs in the conclusion section describes few open problems.
6. The difference between your proposal and related works is not clear, you could to details better. I suggest add a comparative table in ''Related Literature'' to contrast your solution in front of related works. You have added too many references but because they are not in comparison form they seems boring to read.
>> Given the survey nature of this paper and the fact that we do not provide any quantitative improvement in the security bound for any construction, a comparative table seems difficult to accommodate.
7. You could also discuss the relationship between your solution and past literature
>> As mentioned in the response above, most of the results do not provide ant quantitative improvement when compared to the previous results. They are mostly rewritten in H-technique, to first demonstrate the utility of the tool, and second and more importantly demonstrate the proof simplicity in some cases.
Reviewer 2 Report
In this systematization of knowledge (SoK) paper, the authors aim to provide a brief survey on the H-technique. The SoK is in four parts: First, they redevelop the necessary nomenclatures and tools required to study the security of any symmetric key design, especially in the H-technique setting. Second, they give a full description of the H-technique and some related tools. Third, they give (simple) H-technique-based proofs for some popular symmetric-key designs, across different paradigms. Finally, the authors show that H-technique can provide optimal bounds on distinguishing advantages. As far as I know, both the scheme and the proofs are correct. Indeed, I found the idea is interesting. Thus, I can recommend this manuscript for publication in Entropy.
Author Response
Not applicable.